# Immersive scene representation in human visual cortex with ultra-wide-angle neuroimaging

Jeongho Park [1] ✉, Edward Soucy[2], Jennifer Segawa[2], Ross Mair [2,3,4] & Talia Konkle [1,2,5]

While human vision spans 220°, traditional functional MRI setups display images only up to central 10-15°. Thus, it remains unknown how the brain represents a scene perceived across the full visual field. Here, we introduce a method for ultra-wide angle display and probe signatures of immersive scene representation. An unobstructed view of 175° is achieved by bouncing the projected image off angled-mirrors onto a custom-built curved screen. To avoid perceptual distortion, scenes are created with wide field-of-view from custom virtual environments. We find that immersive scene representation drives medial cortex with far-peripheral preferences, but shows minimal modulation in classic scene regions. Further, scene and face-selective regions maintain their content preferences even with extreme far-periphery stimulation, highlighting that not all far-peripheral information is automatically integrated into scene regions computations. This work provides clarifying evidence on content vs. peripheral preferences in scene representation and opens new avenues to research immersive vision.

When we look at the world, we feel immersed in a broader visual environment. For example, the experience of a view of an expansive vista from the top of a mountain is not the same as when looking at a picture of the same view. One key difference is that in the real world, we sense a >180 degrees view of the environment at each glance. Indeed, while our fovea and macula ensure high-resolution input at the center of gaze, there is an equally impressive expanse of peripheral vision: with 170 degrees sensed by a single eye, and up to 220 degrees of the extreme far-periphery sensed by the two eyes combined[1]. What are the neural processes by which this immersive visual experience of the broader environment is constructed in the human visual system?

Seminal research has identified three brain regions in the human brain that show a clear role in high-level visual scene perception[2,3]. There are parahippocampal place area (PPA[4]) in the temporo-occipital cortex, retrosplenial cortex (RSC[5]) or medial place area (MPA[6]) in the medial side along the parietal-occipital sulcus, and occipital place area

(OPA[7,8]) in the parieto-occipital cortex. Extensive neuroimaging studies have characterized tuning properties of these regions and their complementary roles in scene perception, regarding recognition[9–13] and navigation[14–20] in particular.

However, the constraints of standard fMRI image projection setup have limited scene perception research to the central 10-20 degrees of the visual field, with scene properties inferred from postcard-like picture perception. Thus, it remains unknown how a scene activates the visual system when it is presented across the full visual field, providing a more immersive first-person view. Would this alter the way we define the scene regions along the cortical surface (e.g., a larger cortical extent, or new scene regions)? More generally, what are the neural processes that construct a visual scene representation when far-peripheral information is available?

Here, drawing inspiration from an infant fMRI study[21], we introduce an innovative image projection setup, which enables the

[1]Department of Psychology, Harvard University, Cambridge, MA, USA. [2]Center for Brain Science, Harvard University, Cambridge, MA, USA. [3]Department of Radiology, Harvard Medical School, Boston, MA, USA. [4]Department of Radiology, Massachusetts General Hospital, Boston, MA, USA. [5]Kempner Institute for Biological and Artificial Intelligence, Harvard University, Boston, MA, USA. ✉e-mail: jpark3@g.harvard.edu

presentation of ultra-wide-angle visual stimuli in an fMRI scanner. In typical scanning setups, stimuli are presented to humans lying supine in the scanner by projecting onto a screen outside of the scanner bore, while the participants look out through a head coil at a small mirror reflecting the screen behind them. With this setup, the maximum visual angle of a projected image is ~15–20 degrees. We modified this setup, by bouncing the projected image off two angled mirrors, directly onto a large, curved screen inside the scanner bore. This allowed us to project images about 175 degrees wide, stimulating almost the entire visual field.

While there have been prior approaches to establish wide-angle presentation, they were mainly centered on studying retinotopic properties in early visual areas, presenting expanding rings and rotating wedges in black and white[22–25]. Thus, for testing high-level visual areas, which involves presenting more complex images (e.g., faces or scenes), different solutions imposed specific limitations. For example, one approach enabled researchers to project images up to 120 degrees, but only to one eye at a time, and onto a screen that was 3 cm from an eye, requiring participants to view stimuli with a custom contact lens[24–29]. More recently, a high-resolution MR-compatible head mounted display was developed, but the maximum field-of-view is ~52 degrees wide (Nordic Neuro Lab). Our solution was developed with the intention of studying high-level visual perception by providing as expansive and natural visual experience as possible. Further, our approach does not require participants to wear additional devices, and leverages a relatively low-tech solution that can be implemented in other scanning facilities.

With this full-field neuroimaging setup, we first chart the cortex with far-peripheral sensitivity. Then, we leverage this wide-angle setup

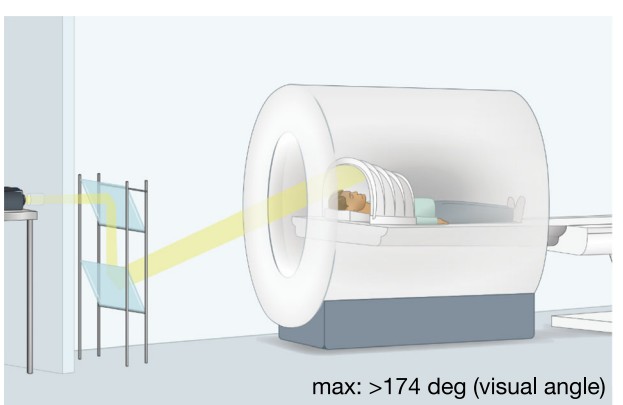

max: >174 deg (visual angle)

**Fig. 1 | Full-field neuroimaging setup. The yellow line represents the projected image trajectory.** An image is bounced off two angled mirrors and directly projected onto a curved screen inside the scanner bore.

to entertain questions about what it means to be a scene and the implications for the responses of classic scene-selective regions. For example, perhaps any image content presented in the far-periphery is part of a scene, and should be automatically integrated into the computations of high-level scene regions. From an embodied, ego-centric perspective, this is a reasonable account. Alternatively, perhaps the scene regions are more like high-level pattern analyzers that are sensitive to particular kinds of image statistics (e.g., open/closed spatial layout, contour junctions, etc.) rather than to the retinotopic location of the visual stimulation per se. Indeed, in the scene perception literature, there is evidence for both accounts. The neuroimaging studies with 0–20 degrees of the visual field showed that the classic scene regions are modulated both by the scene content (over other semantic category contents like faces) and by peripheral stimulation[6,7,30–32]. We now extend the scope of this investigation to the entire visual field and revisit this question.

## Results

### Ultra-wide-angle fMRI

To accomplish ultra-wide-angle visual presentation in the scanning environment, we installed two angled mirrors near the projector such that the projected image was cast directly into the scanner bore, onto a custom-built curved screen positioned around a person's head (Fig. 1, Supplementary Fig. 3). Additionally, given the visual obstruction of the top of the head coil, we simply removed it, allowing participants to have an unobstructed view of the curved screen. Through signal quality check protocols, we confirmed that the lack of top head coil did not have critical impacts on MRI signals for occipital and parietal cortices (see Supplementary Fig. 1 for more details).

To compensate for the curved screen, we developed code to computationally warp any image, to account for the screen curvature and tilted projection angle (Fig. 2). Given the geometrical constraints of our MRI room, only a subset of pixels could be projected onto the screen, resulting in substantially lower image resolution compared to other standard projection systems, particularly in the vertical dimension (see Methods).

Further, we found that when projecting natural scene images across the full field, using standard pictures taken from typical cameras lead to highly distorted perceptions of space—a picture with a compatible wide field-of-view was required. Thus, for the present studies, we built virtual 3D environments in Unity game engine (Unity Technologies, Version 2017.3.0), where we could control the viewport height and field-of-view when rendering scene images. Further details about the full-field fMRI setup can be found in the Methods and on our website (https://jpark203.github.io/fullfield-neuroimaging). Taken together, our solution enabled us to present images over 175 degrees, providing natural and immersive viewing experience.

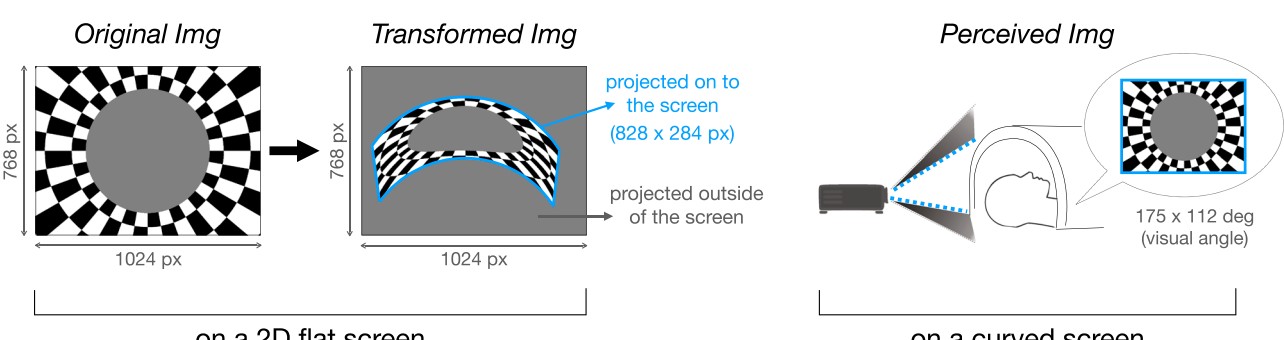

**Fig. 2 | Image projection.** A rectangular image (1024 × 768 pixels) was computationally warped to match the size and curvature of the tilted screen. Due to the geometrical constraints of the room, only a subset of pixels could be projected onto the screen (828 × 284 pixels). On the curved screen, the aspect ratio of the original image was maintained on the display surface.

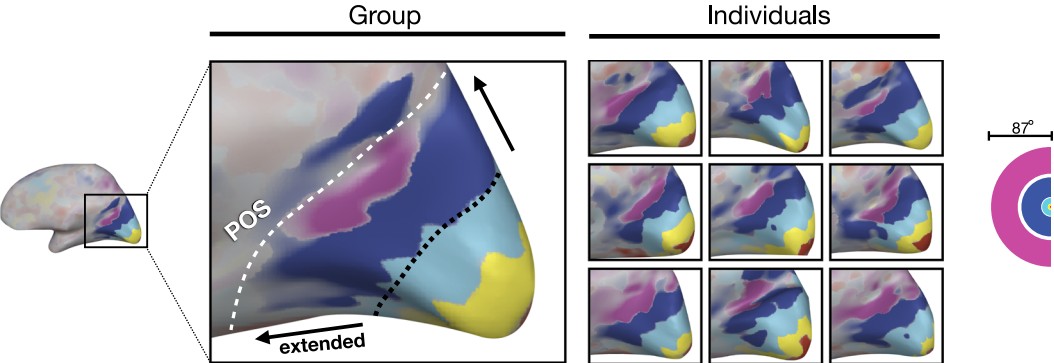

**Fig. 3 | Extended eccentricity map.** An example participant's right occipital cortex is shown from a medial view. Each voxel is colored based on its preference for one of five eccentricity rings (right). In the group data, the black dotted line shows where a typical eccentricity map would end, and the black arrows show how much more the cortex can be stimulated with full-field neuroimaging. Individual brain maps from nine participants also show a consistent pattern of results. POS parieto-occipital sulcus.

## Full-field eccentricity map

In Experiment 1, we first attempted to map the full visual field and chart an extended eccentricity map along the visual cortex. We used a classic retinotopic mapping protocol, where participants performed a fixation dot color detection task. Flashing checkerboards were presented in rings at five levels of eccentricity: (1) a center circle of 1.8 degrees radius, and (2) the inner and outer rings of 2.0–5.6 degrees, (3) 6.3–16.5 degrees, (4) 18.5–50.3 degrees, and (5) >55.3 degrees radius. The two farthest eccentricities were not possible with typical scanning setups, allowing us to stimulate cortical territory that has been inaccessible via direct visual input.

The cortical map of eccentricity preferences is shown in Fig. 3. For each voxel, we compared responses to different eccentricity conditions, and colored the voxel based on the condition with the highest activation (hue). The resulting map revealed a systematic progression of preference from the center to far-periphery, covering an expansive cortical territory along the medial surface of the occipital lobe. In particular, we mapped strong responses to far-peripheral stimulation near the parieto-occipital sulcus (POS), extending beyond our typical eccentricity band maps (black dotted line, Fig. 3). These results validate the technical feasibility of our ultra-wide-angle projection method, and to our knowledge, show the full-field mapping of eccentricity in the human brain that exceeds the scope of prior studies.

## Full-field scene perception

With this full-field neuroimaging set up, we next measured visual system responses to ultra-wide-angle, immersive real-world scenes and compared them to responses from visually-smaller postcard scenes and unstructured image-statistical counterparts.

Specifically, we created four different stimulus conditions that varied in presentation size (full-field vs. postcard), and content (intact vs. phase-scrambled scenes). The full-field images filled up the entire screen (175 deg wide), and the postcard images were presented at the center of screen in a much smaller size (though still 44 deg wide). The chosen size of postcard images was bigger than the maximum size in typical fMRI setups due to limited image resolution. We discuss this limitation further in the Discussion.

To match the image content across presentation sizes, the postcard images were rescaled from the entire full-field images, instead of cropping the center only. To vary the image content, the same scenes were phase-scrambled, preserving the summed spatial frequency energy across the whole image but disrupting all second-order and higher-level image statistics present in scenes[33,34]. Additionally, we also included a postcard-face condition where a single face was presented at the center of screen, in a similar visual size to the postcard-scenes. Each stimulus condition was presented in a standard blocked design

(12 sec), and participants performed a one-back repetition detection task (see Methods for further details).

First, we asked how the visual cortex responds to full-field size images with intact scene content, compared to images with phase-scrambled scene statistics (Fig. 4a). This contrast is matched in full-field retinotopic footprint, but different in the image content. Will the immersive full-field scenes recruit additional brain regions, e.g., with more extensive scene regions (in terms of cortical surface area), or separate brain areas away from the classic scene regions that were not found with the traditional fMRI setups due to the limited stimulus size?

The whole-brain contrast map is shown with the group data in Fig. 4a (Supplementary Fig. 4 for individual participants). We found qualitatively higher responses for intact scenes over the scrambled scenes along the ventral medial cortex, as well as dorsal occipito-parietal cortex. For comparison, we defined three scene ROIs by contrasting the postcard-scene vs. postcard-face condition, reflecting a more typical (non-full field) definition of these regions. Fig. 4a shows the overlaid outlines of these classically-defined ROIs (PPA; OPA; RSC). Note that these ROIs reflect group-level ROIs for visualization, but all ROIs were defined in individual subjects in independent data. Qualitative inspection reveals that these ROIs largely encircle the strongest areas of scene-vs-scrambled response preferences. In other words, it is not the case that the full-field stimulation leads to strong scene content-preferring responses that clearly extend well beyond the postcard-defined ROI boundaries.

One important note is that our postcard-sized stimulus was still rather large (44 degrees) relative to the visual size presented in typical set ups (15-20 degrees). Thus, the present data indicate only that the extent of activated cortical surface is not much increased by a relatively dramatic stimulus size increase from 44 to 175 deg. If there is increasing cortical scene-selective territory as a function of visual angle, it is limited to visual size increases from 15-44 degrees. More detailed parametric visual size mapping is required to answer this question. For the purposes of the present work, these results reveal that the standard contrasts for defining classic scene regions reflect stable functionally defined regions, across both our postcard and full-field presentation sizes.

Next, we asked how the visual cortex responds to full-field scenes compared to postcard scenes. This contrast is matched in content (i.e., identical scene images that have been rescaled), but different in retinotopic footprint (Fig. 4b). This allows us to examine which cortical territory is more active under an immersive visual experience of a scene view, compared to postcard scene perception.

A whole-brain contrast map is shown in Fig. 4b (Supplementary Fig. 5 for individual participants). This map shows that cortex near the POS is activated significantly more to full-field scenes than postcard

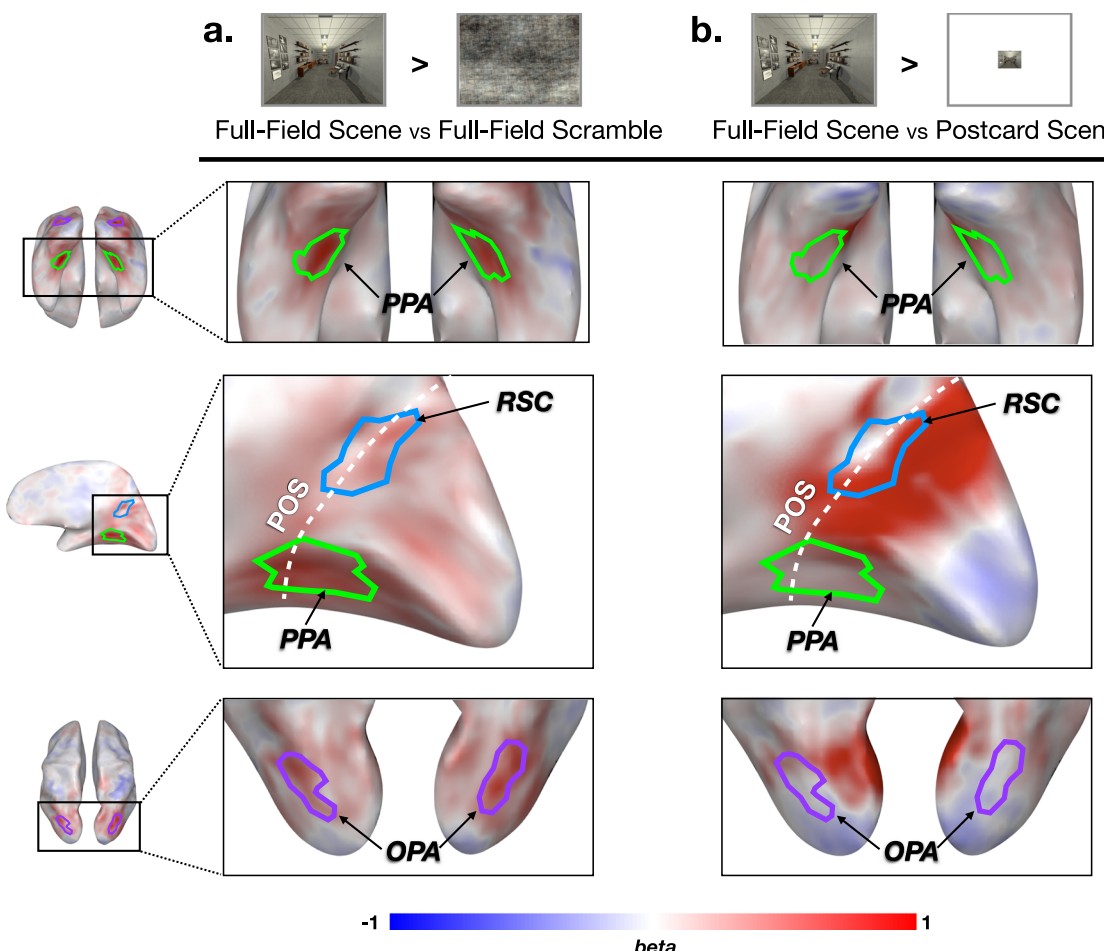

**Fig. 4 | Whole-brain contrast maps.** The group data is shown on an example subject brain. Zoom-in views at each row are captured around the classic scene regions. **a** Image content contrast. A large portion of high-level visual areas, including the scene regions, shows higher activation for the intact scenes compared to the phase-scrambled scenes. **b** Visual size contrast. A large swath of cortex near the parieto-occipital sulcus is strongly activated when viewing a full-field scene compared to a postcard scene. PPA parahippocampal place area, RSC retrosplenial cortex, OPA occipital place area, POS parieto-occipital sulcus.

scenes. This cortex showed far-peripheral visual field preference in Experiment 1, and effectively corresponds to the far-peripheral parts of early visual areas. Thus, it is likely that this cortex is not uniquely attributed to scene content presentation per se, but to any far-peripheral visual stimulation (which we explore further in the next experiments). Anatomically, this swath of cortex is largely adjacent to and mostly non-overlapping with classic scene regions, PPA and OPA, and anterior part of RSC. Thus, while it could have been that the full-field vs. postcard contrast would strongly encompass the scene-selective regions, this was not the case.

### Effects of visual size and scene content

The whole-brain contrasts did not show clear evidence for a new scene region, or more extensively activated cortical surface area from the classic scene regions. Thus, we focused our quantitative analyses on these classic scene ROIs defined at the postcard visual size, and explored the extent to which each scene region is modulated by the visual size and scene content.

In addition to the scene ROIs, we defined a "Peripheral-POS" (parietal-occipital sulcus) region, using the retinotopy protocol data from Experiment 1. Specifically, we selected voxels that survived a conjunction contrast between pairs of the far-peripheral eccentricity ring condition and all other eccentricity ring conditions. Further, we removed the small proportion of voxels in the Peripheral-POS which spatially overlapped with independently defined RSC (mean = 5.6%, std = 7.7% of Peripheral-POS voxels).

The results of the ROI analyses are shown in Fig. 5. Broadly, this $2 \times 2$ design reveals a powerful transition in the invariances of the responses, from cortex with retinotopic selectivities to scene content selectivities. Specifically, the Peripheral-POS region showed clear retinotopic modulation: there was a large effect of full-field vs. postcard sizes ($F(1, 36) = 518.6$, $p < 0.01$, etaSq = 0.91), with only weak effect of image content ($F(1, 36) = 11.7$, $p < 0.01$, etaSq = 0.02), and no interaction between these factors ($F(1, 36) = 1.8$, $p = 0.2$). Put succinctly, this region shows clear retinotopic modulation, with little sensitivity to higher-order scene image content.

In contrast, both the PPA and the OPA showed the opposite pattern. That is, there were large effects of scene content vs. scrambled content (PPA: $F(1, 36) = 535.2$, $p < 0.01$, etaSq = 0.86; OPA: $F(1, 36) = 168.9$, $p < 0.01$, etaSq = 0.8), with only small effects of image size (PPA: $F(1, 36) = 44.7$, $p < 0.01$, etaSq = 0.07; OPA: $F(1, 36) = 5.1$, $p < 0.05$, etaSq = 0.02). There was a very small interaction of these factors in PPA, but not in OPA, with slightly higher activation in PPA for scenes in full-field presentation (PPA: $F(1, 36) = 6.5$, $p < 0.05$, etaSq = 0.01; OPA: $F(1, 36) = 0.6$, n.s.). Thus, intact scenes drive much higher response than the phase-scrambled scenes in PPA and OPA, generally independently of the presentation size (darker vs. lighter color bars, Fig. 5).

The RSC immediately abuts the Peripheral-POS region. Interestingly, it has a slightly more intermediate pattern, though it is more like the other high-level scene regions. That is, RSC showed a large effect of scene content (RSC: $F(1, 32) = 141.1$, $p < 0.01$, etaSq = 0.52) and a moderate effect of visual size (RSC: $F(1, 32) = 93.1$, $p < 0.01$, etaSq = 0.34),

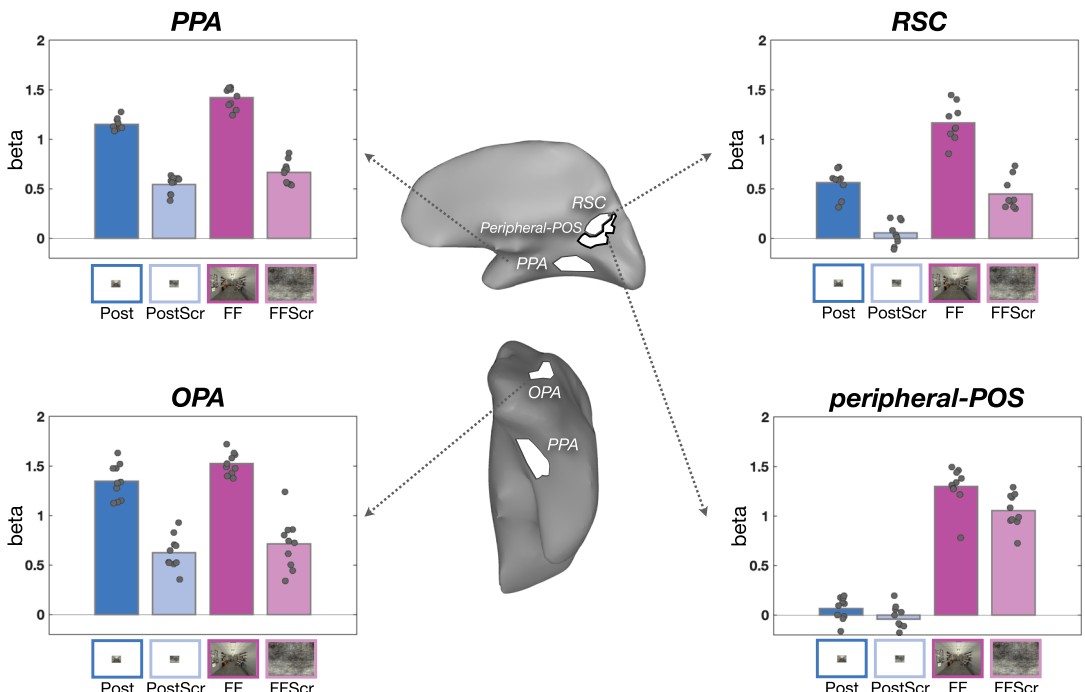

**Fig. 5 | ROI anlaysis.** The anatomical locations of each ROI are illustrated on a schematic brain map in the middle (top: medial side, bottom: ventral surface of the right hemisphere). Each ROI panel shows the mean beta averaged across participants ($n = 10$) for each condition. Individual data are overlaid on top of the bars as dots. The main effect of visual size (blue vs. purple) and the main effect of content (dark vs. light) were significant in all ROIs. The significant interaction was found only in the PPA and RSC. The FFA result is in Supplement Fig. 6. Post Postcard, PostScr Postcard Scrambled, FF full-field scenes, FFscr full-field scenes scrambled, PPA parahippocampal place area, RSC retrosplenial cortex, OPA occipital place area, FFA fusiform face area, POS parieto-occipital sulcus.

with only very weak interaction between them (RSC: $F_{(1, 32)} = 4.3$, $p < 0.05$, etaSq = 0.02). Taken together, these data reveal a clear pattern: classic scene regions have strong overall responses for image content, which is maintained over dramatically different visual sizes and a qualitatively different immersive experience, with relatively weaker modulation by the visual size of stimulus.

As a control, we also examined responses in the face-selective FFA (Supplementary Fig. 6). While the overall responses to all four conditions were quite low, there was a small but statistically reliable main effect of visual size, with higher overall activation in full-field over postcard views ($F_{(1, 36)} = 8.9$, $p < 0.01$, etqSq = 0.19). The responses of this control region suggest that full-field stimulation might partly provide a more general boost to the visual system (e.g., via arousal). On this account, the scene regions' slight preference for full-field stimulation might reflect a more general drive, further amplifying the dissociation between tuning for content and peripheral stimulation.

Thus, from the far-peripheral retinotopic cortex to the classic scene regions, there is a relatively abrupt transition in tuning along the cortical sheet. The far-peripheral retinotopic cortex shows only weak content differences. Adjacent scene-selective cortex amplifies these scene vs. scrambled content differences, regardless of whether or not the content stimulates the far periphery.

### Far-peripheral stimulation without the central visual field
The previous experiment showed that scene regions are modulated dominantly by the image content, much less so by the visual size. However, postcard and full-field scenes both stimulate the central 45 degrees of the visual field. Thus, it is possible that the scene content preferences we observed are actually primarily due to central visual field stimulation. Are these scene content preferences also evident when only stimulating the far-periphery? In Experiment 3, we asked how far in eccentricity this scene preference is maintained.

We also asked the parallel question for face-selective regions. FFA is traditionally defined by contrasting responses to face vs. object

image content presented in the center of the visual field. What happens when faces are presented in the far-periphery? Do face-selective regions also maintain their face content preferences when only presenting the content in the very far-peripheral visual field? Or, will any structured image content be represented increasingly more like a "scene" and drive scene regions, as it is presented farther from the center?

To directly test these questions, we generated a new stimulus set, depicting different content across the visual field, with increasing degrees of central "scotoma"[35], that have matched retinotopic footprint to full-field scenes but differ in their content (Fig. 6). As in the previous experiment, we included both wide-angle rendered 3D scenes and their phase-scrambled counterparts. As a proxy for "full-field faces", we made face arrays, in which multiple individual faces were presented throughout the full visual field. To avoid crowding effect and make each face recognizable (at basic category level), we adjusted the size of faces as a function of eccentricity (see Methods). Object arrays were generated in the same manner with individual small objects.

Then, we parametrically masked the central portion of images at 5 sizes (0, 30, 58, 88, and 138 degrees in diameter; see Fig. 6). We measured brain responses to these 20 conditions, using a blocked design (see Methods). Participants were asked to perform a one-back repetition detection task while fixating their eyes at the center of screen. As before, we defined the classic scene ROIs using the same method (i.e., postcard-scene vs. postcard-face) from independent localizer runs.

We first examined responses of scene and face ROIs (Fig. 7). As expected, when there is no scotoma, all regions showed preferences for either scenes or faces relative to other categories. As the size of the central scotoma increases, leaving only increasingly peripheral stimulation, the results showed that content preferences across all ROIs were generally maintained. Through the penultimate scotoma condition (88 deg), all scene regions showed significantly higher activation for scenes compared to face arrays, object arrays, and

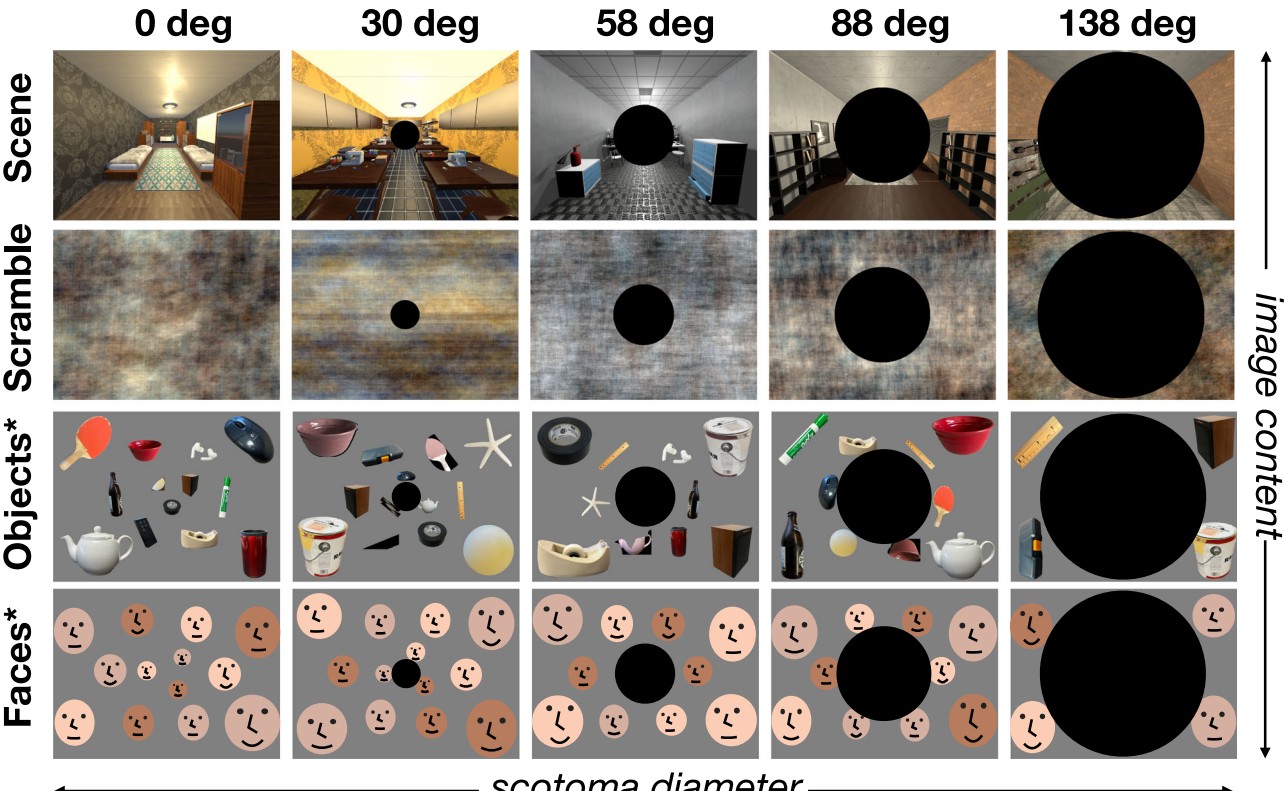

**Fig. 6 | Conditions and stimuli (experiment 3).** To stimulate only the peripheral visual field, we removed the central portion of the image by creating an artificial scotoma that systematically varied in size. There were five levels of scotomas including the no-scotoma condition (columns). We filled in the remaining space with four different kinds of image content: intact scenes, phase-scrambled scenes, object array, and face arrays (rows). For the object and face arrays, the size of individual items was adjusted to account for cortical magnification. *For copyright reasons, human faces have been substituted with illustrations in this manuscript, and objects were substituted with example images without copyright.

phase-scrambled scenes (see Supplementary Tables for statistical test results).

The pattern at the farthest scotoma condition (138 deg) varied by the ROI and stimulus. RSC showed strong scene preference against all other image contents (Fig. 7b, Supplementary Table. 2). However, OPA's scene preference did not hold at the 138 deg scotoma condition (Fig. 7c, Supplementary Table. 3). The PPA showed significantly higher activation for scenes compared to face arrays, but this activation level was not different from object arrays ($t(9) = 2.2$, n.s.; Figure 7a; Supplementary Table. 1). These results are also depicted on the cortical surface in Fig. 8 (Supplementary Fig. 7 for individual participants), showing the contrast of face vs. scene content, as the presentation is restricted increasingly peripherally. Overall, our results show that scene regions can be driven by content differences through a purely peripheral route, beyond at least 88 deg, that does not require central presentation.

Next we turned to FFA. If the presence of faces at the central visual field is necessary to drive FFA responses, then we would have expected the face preference to exist only in the no-scotoma or small scotoma conditions. However, that is not what we found. Instead, face-selective FFA shows the same pattern as the scene-selective regions. That is, FFA responded more to face content than other image content, across all scotoma levels, even at 138 degrees (see Supplementary Table. 4 for stats). This pattern of results is also evident in the cortical maps of Fig. 8 (Supplementary Fig. 7 for individual participants). Overall, these results clearly demonstrate that face-selectivity is present even when faces are presented in the very far periphery only. Thus, this result suggests that there is also a far-peripheral route to drive face-selective responses in the FFA, which does not require direct stimulation of the central visual field.

Finally, we wondered whether participants would actually be aware of the stimulus condition when it was presented in the far 138+ degrees of the visual field. To explore this, we conducted a brief categorization test during the anatomical scan. Either an object array or face array was presented with one of four scotoma sizes, and participants did a 2-alternative-forced-choice task. We found that participants were nearly perfect through the penultimate scotoma condition (30 deg: mean = 0.98, s.e = 0.02; 58 deg: mean = 0.96, s.e. = 0.03; 88 deg: mean = 0.99, s.e. = 0.01). The accuracy at the farthest eccentricity was more variable, but still statistically above chance (mean = 0.64, s.e. = 0.04; $t(11) = 4.0$, $p < 0.01$). We note that only a limited number of trials were conducted due to time constraints, so these results should be interpreted with caution. However, the current results suggest that participants, on average, were weakly able to do the basic-level categorization, with only extreme peripheral visual information present.

**Peripheral bias in scene regions**

Lastly, in the classic scene regions, we found only minimally higher activation for full-field scenes relative to postcard scenes. Is this finding at odds with previously reported "peripheral bias"? Previous studies indicating a peripheral bias have shown increased activation in the PPA when the stimulated location moves from the central visual field to the periphery, up to 20 deg in diameter[30,36]. Two points are worth clarifying. First, our comparison between full-field scenes vs. postcard scenes is not actually a direct test of central vs. peripheral tuning, as both of these conditions stimulate the central visual field. Second, how much a region is activated depends on its receptive field (RF) size and location. So, for example, if a region's RF completely encompasses the 44 deg diameter center of the visual field (i.e., postcard scene presentation size), that

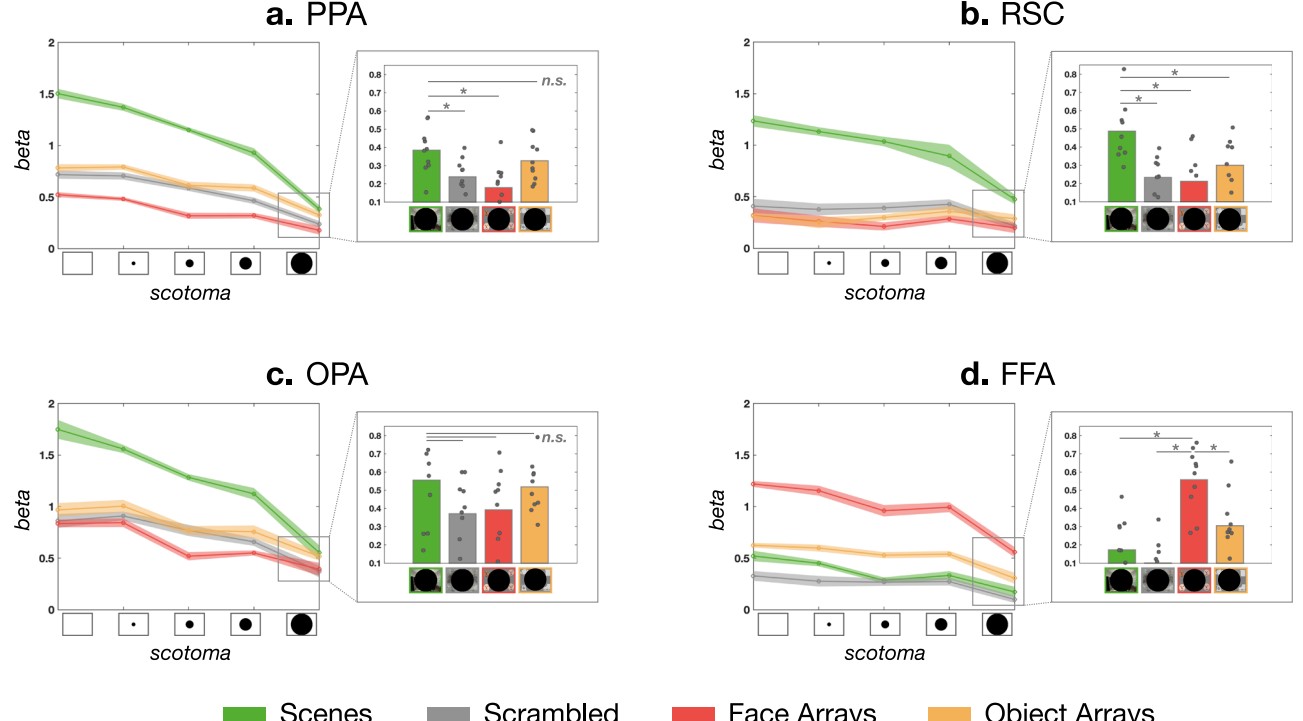

**Fig. 7 | Region-of-interest (ROI) analysis (experiment 3, _n_ = 10).** In each panel, the line plot (error bands = standard error of the mean) shows how the response of each ROI changed as we increasingly removed the central visual field stimulation via scotoma, leaving only the peripheral stimulation. The call-out box with a bar plot (*_p_ < 0.05, two-sided paired t-test; see Supplement Tables for the full report of the statistical tests) shows responses for each image content at the largest scotoma condition (138 deg diameter). **a**, **b** Overall, PPA and RSC maintained their scene preference over faces across all scotoma conditions, whereas **c** the OPA maintained the preference until the penultimate condition. **d** The FFA also maintained its content preference for faces across all scotoma conditions. PPA parahippocampal place area, RSC retrosplenial cortex, OPA occipital place area, FFA fusiform face area.

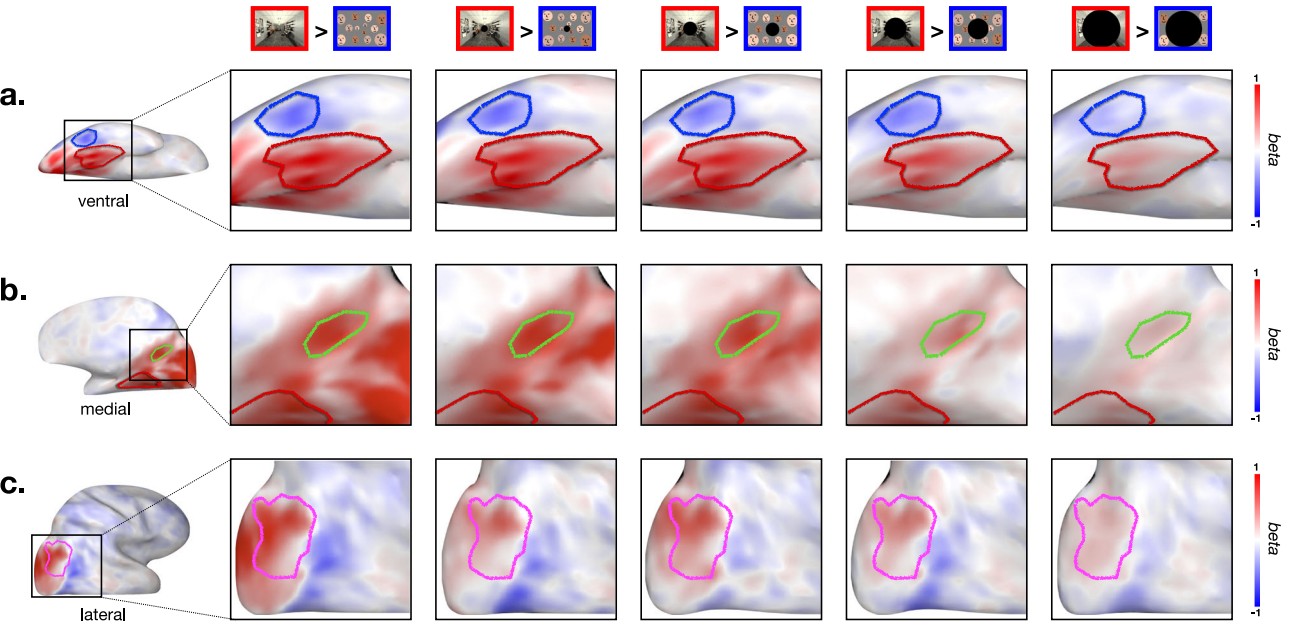

**Fig. 8 | Whole-brain contrast maps (experiment 3).** This figure shows the whole-brain contrast between the scenes (red) and faces (blue), at each scotoma condition (columns). **a** Ventral view with PPA and FFA. **b** Medial view with RSC. **c** Lateral view with OPA. PPA parahippocampal place area, RSC retrosplenial cortex, OPA occipital place area, FFA fusiform face area.

means this brain region's RF would be stimulated in both postcard and full-field scenes, predicting not much activation difference.

We thus ran an exploratory analysis that examined each ROI's response to the increasing eccentricity ring checkerboards used in Experiment 1. A peripheral bias account would intuitively predict that increasing peripheral stimulation would lead to a corresponding activation increase in each of these scene regions. However, that is not what we found. Instead, each scene ROI

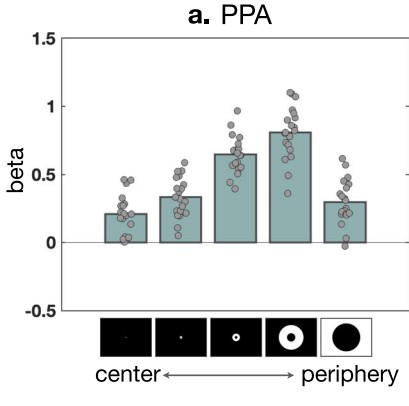

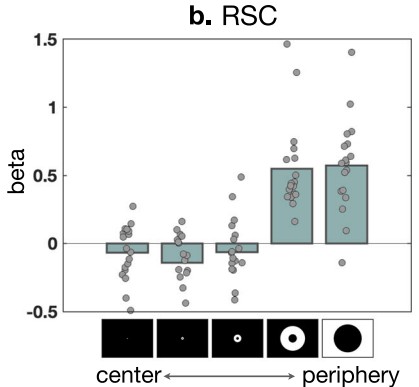

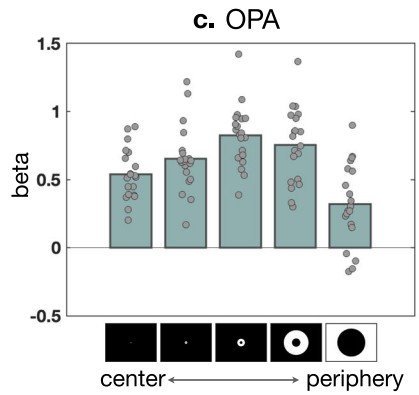

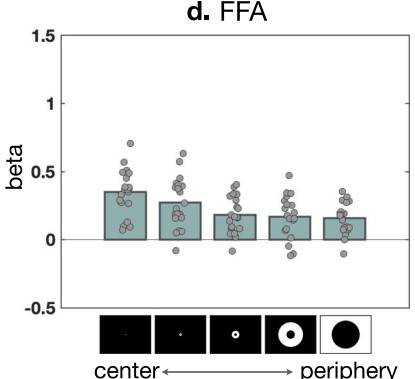

**Fig. 9 | Region-of-interest (ROI) responses to eccentricity rings ($n = 20$). a** PPA response increases until the penultimate condition then drops at the extreme periphery. **b** RSC response was rather flat then jumped after the third ring, clearly showing its preference for the far-periphery. **c** OPA showed a mild peak around the third ring. **d** FFA showed the opposite pattern to **a–c**, demonstrating its preference for the central visual field. PPA parahippocampal place area, RSC retrosplenial cortex, OPA occipital place area, FFA fusiform face area.

showed a different pattern in response to these eccentricity rings (Fig. 9).

The PPA had increasing activation with increasingly peripheral eccentricity rings (up to 37–100.6 deg diameter) but dropped at the farthest, most peripheral stimulation condition (>110 degrees). The OPA was similar to PPA, but with a nominal peak activation at the 3rd level of eccentricity (12.6–33 deg). Finally, RSC's activation to central checkerboards was not significantly different from baseline for the first three levels, and then abruptly increased for both the two most extreme peripheral rings. Thus, neither PPA nor OPA showed strong sensitivity to ultra-peripheral generic stimulation (flashing checkerboard), showing a limit on the general peripheral bias hypothesis of scene regions.

Are these ROI responses across levels of eccentricity consistent with the visual size effects between full-field and postcard conditions? The size of the postcard scene (44 deg diameter) is most similar to the size of the inner circle at the fourth eccentricity ring (37 deg). So, in a rudimentary way, the stimulated visual field by both the last two eccentricity rings (>37 deg) roughly corresponds to the additionally stimulated visual field by the full-field scenes compared to the postcard scenes (>44 deg). Both PPA and OPA have stronger responses for the first three levels of eccentricity than the final two levels, and consistently showed little additional response to full-field scenes relative to postcard scenes. Meanwhile, RSC shows weaker responses for the first three levels of eccentricity, and more for the most peripheral conditions; and consistently, RSC showed stronger responses for full-field conditions regardless of content. Thus, the activation differences over eccentricity rings are indeed consistent with the visual size modulation effect of each scene region, observed in Experiment 2.

In sum, this post-hoc analysis is consistent with the previously known notion that peripheral bias—peripheral stimulation activates the scene regions more than the foveal stimulation. However, our results also place updated constraints on this account. The peripheral bias in the scene regions is present only up to a certain eccentricity, and this differs depending on each scene region. We offer that the thinking of a general peripheral bias is thus not appropriate, and the responsiveness over the visual field might be better understood in the context of RFs. Future work employing population RF mapping can be used to further clarify and chart the far-peripheral RF structure across these cortical regions.

## Discussion

In this study, we established a method to present ultra-wide-angle visual stimuli in the scanning environment. With this new tool, we were able to measure neural responses to the extreme far-periphery and chart the ultra-wide eccentricity map in the human brain beyond the scope of prior studies. We then examined the neural basis of full-field scene perception. We found that classic scene regions are tuned to scene content that is robust to changes in the visual size of scenes, suggesting a sharp tuning transition from adjacent far-peripheral retinotopic cortex to scene content regions. We also found scene and face-selective regions maintained their content preferences even in conditions of extreme peripheral stimulation, highlighting the existence of a far-peripheral route that has yet to be fully investigated. Finally, only RSC showed systematically higher responses at the farthest eccentricity, where both PPA and OPA had weaker responses, clarifying new limits on the peripheral bias of scene regions. Broadly, this work brings unique empirical evidence to clarify debates about the issues of content and peripheral preferences in scene representation

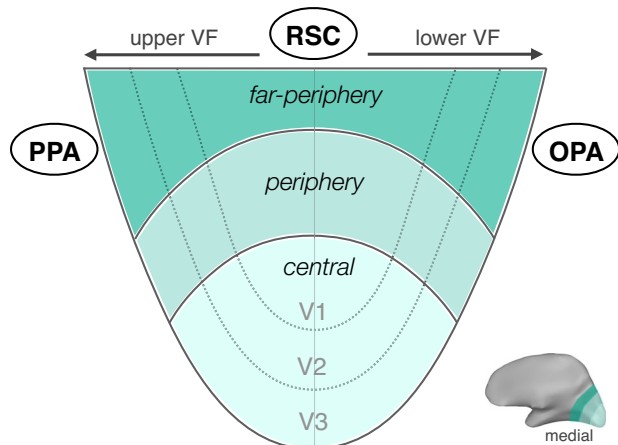

**Fig. 10 | Schematics showing the relationship between the retinotopic map and the scene regions.** The scale and shape of retinotopic map is not accurately presented as the actual data. Instead, this flattened map of the medial view emphasizes the idea that the three scene regions might be connected via the far-peripheral cortex. VF visual field, PPA parahippocampal place area, RSC retrosplenial cortex, OPA occipital place area.

and introduces an innovative method for investigating more naturalistic, immersive scene perception inside a scanner.

The full-field neuroimaging method allowed us to gain some new insights into the classic scene regions. First, we gained a better understanding of what it means to be a scene. While it has been well established that PPA, RSC, and OPA are scene-selective regions, the definition of a scene has been used in a somewhat mixed way. On one hand, a scene can be a set of visual patterns with particular kinds of higher-order image statistics. On the other hand, anything (including objects or faces) that falls in the far periphery can be part of a scene. This account is motivated by intuitions that part of what it means to be a scene is to have content that extends beyond the view. Leveraging the ultra-wide-angle image projection, our study directly compared these two accounts.

Overall results are clearly in favor of the first hypothesis. That is, not just any information in the far-periphery becomes a scene and is automatically integrated into the computations of scene regions. Even when faces or objects are at the far periphery, they do not drive the scene regions more than they would normally do at the central visual field. Instead, the classic scene regions are tuned to particular higher-order image statistics that are distinctive from visual features of other semantic categories, although there are some further differences among the scene regions[2,3]. This makes sense: many of the core visual features important for the scene regions are not much disrupted by the visual size or location change. For example, spatial layout[34,37], statistics of contour junctions[38], surface properties like material or texture[12,39], or objects present in a scene[40] can be extracted similarly in both postcard and full-field scenes. However, it is also worth emphasizing that while these features do not have to be present in specific retinotopic locations, in real visual experience, useful visual cues for those (e.g., walls, planes, or boundaries) tend to be at the periphery rather than the center, providing an ecological explanation why the scene regions are developed to have sensitivity to visual information at the periphery.

Additionally, the access to the far-periphery provided a new perspective on the anatomical locations of the scene regions. We showed that three scene regions are very closely positioned to the far-peripheral cortex along the POS. When we perceive a full-field view, this medial brain region and the classic scene regions are activated together, forming a large ring-shaped portion of the cortex along the POS. In other words, the classic scene regions might be connected

together by the far-periphery preferring cortex. This observation allows us to realize that the scene regions are actually proximal to each other anatomically. This intuition is not easily captured from the typical flattened brain map, because the cut is made along the fundus of calcarine sulcus[41], splitting the retinotopic map into upper and lower visual quadrants, which in turn places PPA and OPA on opposite sides of this flat map (e.g., see ref. 42 for how object-selective cortex is organized between these PPA and OPA regions). Our schematic map of the medial surface (Fig. 10), in contrast, keeps the visual field maps intact, emphasizing the proximity between the scene regions and their relationship to the retinotopic map.

This view naturally lends an explanation why the PPA has upper visual field bias, OPA has lower visual field bias, and RSC does not show clear bias to either upper or lower visual field[31]. Further, this large-scale cortical organization may be related to recently proposed place-memory areas that are positioned immediately anterior to each of the scene-perception areas[43]. In particular, the organization is suggestive of a hierarchical representational motif, with systematic transformations of representation from retinotopic far-peripheral cortex to perceptual scene structure of the current view to more abstract scene memory.

Another interesting question is the relationship between RSC and area prostriata, which is located in the fundus of the calcarine sulcus, just anterior to the far-peripheral V1[29]. The prostriata has a distinctive representation from V1 over a wide visual field (up to 60 deg), and responds more to fast motion (570 deg per sec) than the moderate-speed motion (38 deg per sec)[29]. Moving dorsally along the POS, there is also human V6 that has the sensitivity to coherent field motion (e.g., spiral motion vs. random dots) or optic flow when compatible with self-motion[44,45]. While it requires further investigation whether the prostriata overlaps with the functionally defined RSC, it is possible that its sensitivity to peripheral motion might be used for representing dynamic scenes, to support spatial navigation.

The scene regions and even the fusiform face area both showed their content preference at the extreme far-periphery. How do these regions process stimuli at the far periphery?

Many studies have shown that face-selective regions respond more strongly to foveal stimulation, whereas scene-selective regions respond more strongly to peripheral stimulation[30,36,46]. Further, stronger functional connectivity was found between foveal V1 and face-selective regions (and between peripheral V1 and scene-selective regions), in human adults[47], as well as in human and macaques neonates[48,49]. More recent study using diffusion MRI also showed higher proportion of white matter connection between foveal early visual cortex and ventral face regions (e.g., fusiform face area; FFA)[50]. Together, these results imply eccentricity-based preferential connection between early visual cortex and higher category-selective regions, which does not easily explain our findings.

One possibility is that there are meaningful connections across all eccentricities between the early visual cortex and the higher visual areas, even though some connections to a particular eccentricity are more weighted (e.g., FFA and foveal V1). Then, FFA might still show a somewhat weaker but preferential response to faces at the far periphery, as long as the stimuli are presented with appropriate visual size and arrangement to accommodate cortical magnification and crowding.

Another possibility is that attention temporarily adjusts RF properties of high-level visual areas. A study using the population receptive field (pRF) method showed that the pRFs of FFA were located more peripherally and larger during a face task (one-back judgment) than during a digit judgment task, resulting in extended coverage of the peripheral visual field[51]. While there was no control task condition in our experiments, the one-back repetition detection task could have helped incorporate far-peripheral stimuli into computations.

Additionally, there might be other input connections to the high-level visual areas outside the ventral pathway, perhaps via a subcortical route (e.g., superior colliculus)[52,53] or from the lateral surface. For example, the diffusion MRI study showed that lateral face regions (e.g., posterior STS-faces) have uniformly distributed connections across overall early visual cortex eccentricities, in contrast to the foveal-biased ventral FFA[50]. This suggests that the processing of faces is not limited to the central visual field, as they can also be processed at the periphery, especially in dynamic or social situations[54,55]. It is possible that the peripheral face selectivity observed in FFA may reflect responses from those lateral face areas. Further investigation is necessary to better understand these peripheral routes and how they support the transition from eccentricity-based to content tuning.

Lastly, another possibility to consider is that this effect is driven by non-compliant subjects who moved their eyes to the periphery. However, if participants always shifted their gaze towards the periphery, activation levels at the largest scotoma would match those in the no-scotoma condition, and if such eye movements happened only occasionally, it would likely result in greater variance in the far-periphery condition, which we did not observe. Further, moving your eyes beyond 70 degrees requires considerable effort and some discomfort. Thus we think this account of the responses is unlikely. While the setup here precludes the use of traditional eye-tracking equipment, emerging computational eye-tracking methods that extract eye gaze from the EPI images could prove a valuable complement to this method in future studies[56].

Achieving a wide-angle (>15–20 deg) visual stimulation in an fMRI scanner has been a goal since the early days of fMRI[57,58]. For example, in the early 2000s, researchers were able to stimulate wider visual field up to 100 deg, mapping retinotopic area V6 in humans[22]. To achieve this, a relatively large flat screen (260 × 185 mm) was positioned inside the scanner bore, and the closer distance between the screen and the eyes allowed it to stimulate a larger portion of the visual field. However, this screen size was too large to be easily adaptable to other MRI scanners or conventional head coils. Another research group achieved 100 deg wide stimulation with a smaller screen (140 mm wide), but they used the combination of glasses and prism to enlarge the size of projected stimuli[23,59].

In the next decades, the angle of image projection was pushed up to 120 deg wide[24–29]. These approaches leveraged monocular viewing–presenting the image to only one eye. In these setups, a small screen was positioned very close to the eyes (3 cm), and participants had to wear a contact lens to get help with focus and fixation at such a short distance. And, most recently, stimulation of the entire visual field was achieved[60]. Using custom-built goggles with white light-emitting diodes (LEDs), they were able to functionally localize the temporal monocular crescent, which requires wide-angle projection beyond 120 deg.

While much of this early work focused on retinotopy, our aim was to develop an approach that does not require participants to wear any devices and allows them to see stimuli as naturally as possible as they do outside the scanner. And, we focus here on exploring the perception and representation of high-level visual information presented extensively across the visual field. An advantage of our approach is that apparatus can be built at relatively low cost. We used a pair of mirrors to control the image projection trajectory, and the curved screen can be assembled with 3D-printed plastic parts. We share the design files and all specifications via a public website and community mailing list to support ultra-wide-angle neuroimaging (https://jpark203.github.io/fullfield-neuroimaging).

One of the current challenges of our ultra-wide-angle projection setup is that we are scanning without the top head coil because it blocks the peripheral view. While the data quality was still viable, there was a clear decrease of tSNR in all of the main ROIs (Supplementary Fig. 2). The lack of top head coil could also limit the scope of research topics, especially if they involve investigating on the frontal lobe. Another main challenge is a limited image resolution (2–4 pixels/degree). Due to physical constraints of the scanner room, only ~30% of pixels from the projected image could be on the screen. This is because as the distance between the projector and the screen (inside the scanner bore) gets farther, the size of the projected image also gets larger. However, this limitation in spatial resolution can be overcome with our new projector that supports much higher resolution (up to 4k), compared to the old one (maximum 1024 × 786 pixels), increasing the projected resolution more than threefold (8–15 pixels/degree).

Regardless of these limitations, our full-field scanning method provides promising new research avenues that can be explored in future investigations. One such avenue is to explore how the brain represents the spatial scale of a view in a more ecologically valid manner. Traditionally, we study object-focused views by cropping a picture closely to an object, eliminating all contextual peripheral visual information. However, this picture editing approach does not reflect how we actually experience the world, as we continuously receive visual information from the periphery even when focusing on an object. By simply moving the camera position (as an agent moves in an environment) and maintaining the same wide field-of-view, the spatial scale of the view is naturally determined by the distance between the focused object and the camera (agent). This positions us to investigate how we obtain a sense of object-focused view in the real visual world. Moreover, this method allows us to re-examine previous studies on various aspects of spatial representation in the brain. We can revisit how the continuous dimension of space is represented from an object-focused view to a far-scale navigable scene view[61], how intermediate-scale scenes (e.g., a view of a chopping board) are represented in the brain[62], and how the memory of a view is biased depending on the depicted spatial scale[63,64]. Importantly, this can be done while isolating field-of-view manipulation (e.g., cropping) from viewing distance manipulation.

Another promising research direction is to investigate the role of peripheral vision in computing one's body position (relative to objects or environments) in complex, dynamically moving situations. This task is crucial for activities ranging from maneuvering a vehicle and playing sports to everyday walking and navigation. For this, extracting relevant visual cues such as optic flow, and sensitivity to the peripheral visual field in particular would be important. Notably, brain regions involved in these processes, such as the peripheral POS, human V6, prostriata, and potentially OPA[17,22,29], are spatially adjacent along the POS. Full-field scanning offers a unique opportunity to directly stimulate these regions. This approach can enhance our understanding of how these areas interact and contribute to ego-motion computation, with wide-reaching implications for applied vision research.

The present findings reveal that classic scene regions are modulated by structured image and scene content, over dramatic changes in visual size, suggesting that they are tuned to particular higher-order image statistics rather than to any peripheral stimulation. Broadly, this study demonstrates how full-field neuroimaging allows us to investigate visual perception under more realistic, immersive experiences.

## Methods

### Participants

Twenty-two participants were recruited from the Harvard University Public Study Pool (10 females aged 20–54 years). All participants completed Experiment 1 (retinotopy protocol), ten participants in Experiment 2, and twelve participants in Experiment 3. All participants had normal or corrected-to-normal vision, gave informed consent, and were financially compensated. The experiments were performed in accordance with relevant guidelines and regulations and all procedures were approved by the Harvard University Human Subjects Institutional Review Board.

## Apparatus

To enable ultra-wide-angle projection during scanning, several modifications were made to the typical scanning setup. In order to achieve an unobstructed view for the participant, we did not attach the top head coil and scanned only with the bottom head coil. Instead, we placed a custom-built curved screen right above the participant's head. The screen was built with 3D-printed plastic parts and acrylic solvent. The curved shape was maintained by gluing a polystyrene sheet (1/16 inch thick) to a custom-fabricated acrylic hull (Supplementary Fig. 3b). The radius of the cylindrical screen was 11 inches. The screen was made as large as possible while remaining rigidly supported and still fitting inside the MRI bore (about 12-inch radius). The one-inch difference allowed for the supporting ribs of the hull and a bit of clearance when moving in and out of the bore. Adjustable "legs" were attached at the bottom of the screen with nylon screws, and these legs were slotted into the scanner bed, allowing the screen to be securely anchored. Design files of the screen can be downloaded at https://jpark203.github.io/fullfield-neuroimaging/screen.

We also removed the standard flat projection screen at the back of the scanner bore. We bounced the projected image off of a pair of angled mirrors installed near the projector, directly into this curved screen inside the bore (Supplementary Fig. 3a, Fig. 1). For this, we constructed an inverted periscope. A pair of front surface mirrors were supported on a non-ferromagnetic stand. The lower mirror remains fixed, and the upper mirror is hinged. Tilting the upper mirror up removes the periscope from the projection path. With the periscope in place, the projector appears to originate from a virtual point further back and below the floor of the room.

Since this step changed how far the image on the screen is cast from the projector, we also adjusted the focus setting of the projector. Next, we used a reference image that was warped to fit the screen to check whether the image was accurately projected on the curved screen. If necessary, we carefully adjusted the projector position and/or the mirror angle. After this initial calibration stage, we refined the screen setup after a participant was put inside the scanner. First, we asked the participant to adjust their head position such that they were looking directly toward the center fixation mark on the screen. Second, we further adjusted the focus setup of the projector based on individual participants' feedback. Overall, we allocated ~15–20 minutes of additional time for setting up the full-field scanning.

## Image projection

To increase the spatial extent of stimulus, our goal was to project an image onto the inner wall of the cylinder bore. Ideally, the projector would be incident on the screen at 90 physical degrees. The physical geometry of the scanner bore makes this nearly impossible. The geometry of the room and our projection path are schematized in Supplementary Fig. 3. The next best solution would be to place the projector (or the final mirror) closer to the cylinder bore in order to obtain the steepest angle possible. We did not pursue this route because any alterations must be minimally intrusive to alter any other ongoing study, as the MRI serves many labs.

If we projected directly onto the MRI bore, the light rays would be incident at just over 18 physical degrees. This shallow angle results in large distortion along the vertical (Y) axis of the projected image. To somewhat mitigate this, we angled the projection screen. Rather than being parallel to the magnet bore, we tilted it by 10 physical degrees. The edge of the screen at the top of the subject's head nearly touches the bore. The screen edge near their collarbone is closer to the subject than the bore. Tilting angles larger than 10 physical degrees were ruled out for reasons of comfort—eye strain, feelings of confinement, etc. Effectively, this leads to the projector being angled slightly over 28 physical degrees relative to the screen (i.e., combining the tilted angle of the mirror and the screen).

As a result, approximately 1/3 of the Y pixels of the projector fall onto the screen, limiting our vertical resolution to 284 pixels rather than the native 768. In the case of the x pixels, about 828 pixels fall onto the screen, out of the native 1024 pixels (Fig. 2, Supplementary Fig. 3a). Pixels that did not intercept the display screen were set to black.

The visual angle of the display screen ranges from 168–182 degrees in width and 106–117 degrees in height (Supplementary Fig. 3c). This variation depends on the distance between the participant's eyes and the screen, which is affected by head size and head cushion options, for distances between 13.5–16.5 cm. For the stimulus size to be reported in the manuscript, we picked the middle viewing distance (15 cm) and calculated a stimulus angular extent. Perspective estimates did not take into account subject variability or binocularity.

The resolution of our current screen was 4.6–4.9 pixels per degree in width and 2.4–2.7 pixels/degree in height. It is noteworthy that the current limits on the low resolution can be overcome by our new projector, which has a much higher maximum resolution (4k). For example, if we keep the same aspect ratio of 4:3 (3200 × 2400), the pixels/degree will increase by the scaling factor of 3.125 (i.e., 2400/768 = 3.125).

## Computational image warping

Because of the curvature and angle of the screen, all projected images were first computationally warped using a custom function to compensate for the geometry of the curved screen. Specifically, we developed a computational method that transforms a regular, rectangular image (1024 × 768 pixels; 4:3 aspect ratio) into a curved shape that matches the size and curvature of our custom-built screen. The transformed image on the cylindrical display surface preserved the same original aspect ratio (4:3) as it is measured 58.5 cm (arc length) × 44 cm (linear). Our image-warping algorithm allowed us to project the images onto the cylindrical screen without stretch or distortion; similar to the real-world action of pasting a sheet of wallpaper onto a cylindrical wall.

To link the warping algorithm parameters to the physical set up, we developed a calibration procedure, in which we use an MR-compatible mouse to obtain the × and y coordinates of the projector image that correspond with the three points along the screen outline (e.g., measuring points along both the top and bottom of screen curvature separately, as the bottom screen was slightly narrower than the top). This resulted in a 2d mapping, which takes an original image, and then resizes and warps it to be positioned directly into the part of the projected image that is being projected onto the screen (Fig. 2).

## Signal quality check

Several quality assurance tests were conducted with and without the top head coil separately, to check how much fMRI signal was impacted by removing the top head coil. First, we ran the CoilQA sequence that calculates and provides an Image SNR map. Second, we ran one of our BOLD protocols (i.e., one of the functional runs), computed tSNR maps, and examined BOLD quality check results. Third, we also ran the T1-weighted scan for a qualitative comparison between the two cases. The test results are reported in Supplementary Fig. 1.

Additionally, we also computed tSNR within each ROI. For this, we preprocessed the data using the identical protocol as the main experiments and normalized it into Talairach space. The voxel-wise tSNR was calculated by dividing the mean by the standard deviation of time-course data. Then, we extracted voxels for each ROI, and averaged their tSNRs to get an ROI tSNR value. The comparison between with and without the top head coil is reported in Supplementary Fig. 2.

## Rendering full-field views from virtual 3D environments

Computer-generated (CGI) environments were generated using the Unity video game engine (Unity Technologies, Version 2017.3.0). We constructed twenty indoor environments, reflecting a variety of semantic categories (e.g., kitchens, bedrooms, laboratories, cafeterias,

etc.). All rooms had the same physical dimensions (4 width × 3 height × 6 depth arbitrary units in Unity), with an extended horizontal surface along the back wall, containing a centrally positioned object. Each environment was additionally populated with the kinds of objects typically encountered in those locations, creating naturalistic CGI environments. These environments were also used in refs. [61,63].

Next, for each environment, we rendered an image view, set to mimic the view of an adult standing in a room looking at the object on the back counter/surface. During the development of these protocols, we found that it was important to get the camera parameters related to the field of view (FOV) right to feel as if you were standing in the room with objects having their familiar sizes; otherwise, viewers were prone to experience distortions of space. Here the camera FOV was fixed at 105 degrees in height and 120.2 degrees in width. This FOV was chosen based on the chord angle of our physical screen (120 deg) and empirical testing by experimenters. Since there was no ground truth for the size of virtual reality environments (e.g., how large the space should be), experimenters compared a few different FOVs and made subjective judgments on which parameter feels most natural. Relatedly, we set the camera height to be 1.6 (arbitrary units), and tilted the camera angle down (mean rotation angle = 5.2 deg, s.d. = 0.5 deg, across 20 environments), so that the center object was always at the center of the image. For these stimuli, we positioned the camera at the back of the environment, to give a view of the entire room. Each image was rendered at 1024 × 768 pixels.

## Experiment 1
In the retinotopy runs (5.8 min, 174 TRs), there were 7 conditions: horizontal bands, vertical bands, and five levels of eccentricities (e.g., from foveal stimulation to far-peripheral stimulation). A center circle was 1.8 degrees radius, and the inner and outer rings of the rest of the conditions were 2.0–5.6 degrees, 6.3–16.5 degrees, 18.5–50.3 degrees, and >55.3 degrees radius. All stimuli cycled between states of black-and-white, white-and-black, and randomly colored, at 4Hz. Each run consisted of 7 blocks per condition (6-sec block), with seven 6-sec fixation blocks interleaved throughout the experiment. An additional 6-sec fixation block was added at the beginning and the end of the run. Participants were asked to maintain fixation and press a button when the fixation dot turned blue, which happened at a random time once per block.

## Experiment 2
In Experiment 2, participants completed 8 runs of the main protocol (one participant completed 6, and two participants completed 5 runs) and 3 retinotopy runs (two participants completed 2 runs).

In the main protocol, there were 7 stimulus conditions. (1) Full-field scenes: 15 full-field scene images were chosen (randomly selected from the 20 total environments). (2) Full-field Phase-scrambled image. First, the images were fast Fourier transformed (FFT) to decompose them into amplitude and phase spectrum. Then, the phase spectrum was randomized by adding random values to the original phase spectrum. The resulting phase spectrum was combined with the amplitude spectrum, then transformed back to an image using an inverse Fourier transform[65]. (3) Postcard scenes. These images were generated by rescaling the full-field scenes. Instead of cropping the central portion of the original image, an entire image was rescaled from 1024 × 786 pixels to 205 × 154 pixels (44 degrees wide). This rescaled image was positioned at the center, and the rest of area around it was filled with the background color, such that the size of whole image (i.e., small scene at the center with the padding around it) was kept the same as the original image (1024 × 768 pixels). (4) Postcard-scrambled scenes. The same rescaling procedure was followed for the phase-scrambled scenes. The final three conditions consisted of fifteen images from each category of (5) faces, (6) big animate objects, and (7) small inanimate objects. They were rescaled to fit a bounding box (171 × 129 pixels; 37 degrees wide) with white background color. This bounding box was positioned at the center with the padding, so that the size of an output image is 1024 × 768 pixels.

A single run of the main protocol was 6.5 min in duration (195 TRs) and was a classic on-off blocked design. A condition block was 12 sec, and was always followed by 6 sec fixation period. Within each block, six trials from one condition were presented. Each trial consisted of 1.5 sec stimulus presentation and 500 ms blank screen. The stimulus duration was chosen to be a little longer than the typical scanning, because flashing full-field images too fast can be uncomfortable and may cause nausea. Among those six images, five were unique images, and one of those images was randomly chosen and repeated twice in a row. Participants were instructed to press a button when they saw the repeated image (one-back repetition detection task). The presentation order of blocks was pseudo-randomized for each run as follows. Seven conditions within an epoch were randomized 3 times independently and concatenated with a constraint that the same condition cannot appear in two successive blocks. Thus, each of 7 condition blocks were presented 3 times per run. Fifteen unique images per condition were randomly split across those three blocks, for each run.

## Experiment 3
In Experiment 3, participants completed 8 runs of the main protocol (one participant completed 7, and five participants completed 6 runs), 2 runs of classic category localizer (two participants completed 1 run, and two participants did not complete any localizers and were excluded from ROI analyses), and 2 retinotopy runs (two participants completed 3 runs).

In the main protocol of Experiment 3, stimuli were varied with 2 factors: image content (scenes, phase-scrambled scenes, face arrays, object arrays), and scotoma size (0, 29, 58, 88, 140 degrees in diameter). The scene images were captured from 20 virtual environments built in Unity, using the same camera parameters as in Experiment 2. For face and object conditions, 58 individual faces and objects were collected. We matched the luminance across scenes, faces, and objects, by equating the luminance histograms using Color SHINE toolbox[66]. The phase-scrambled scenes were generated from the luminance-matched scenes, using the same parameters as in Experiment 1.

Face and object arrays were generated with those luminance-matched images. For each face array, 13 faces were randomly drawn from a pool of 58 faces (half male). These faces were arranged along 3 levels of eccentricity circles. The size of individual faces and the number of faces was adjusted for each eccentricity, to account for cortical magnification and to avoid crowding effect. At the smallest eccentricity, 3 faces were rescaled to the size of 113-pixel diameter; at the middle eccentricity, 6 faces were rescaled to the size of 178-pixel diameter; at the largest eccentricity, 4 faces were rescaled to the size of 295-pixel diameter. The largest faces were positioned at 4 corners of the image, and the rest of faces were equally distanced along the eccentricity circle, with random jitters applied to individual face locations. Object arrays were generated using the same procedure. This step resulted in 20 face arrays and 20 object arrays. After making those base stimuli with 4 different image content (scenes, phase-scrambled scenes, face arrays, object arrays), we generated scotoma conditions by applying scotoma masks with 5 levels: 0 (i.e., no mask), 29, 58, 88, and 140 degrees in diameter. In total, 400 unique stimuli were generated across 20 conditions.

The main protocol was 6.9 min in duration (208 TRs), and used a block design, with 20 conditions presented twice per run. In each condition block (8 sec), five trials from one condition were presented. Each trial consisted of 1.1 sec stimulus presentation, followed by 500 ms blank screen. A fixation (black and white bullseye) was presented at the center of screen throughout an entire block. Among those five images in a block, four were unique images, and one of those images was randomly chosen and repeated twice in a row. Participants were asked to press a button when they detected the repetition. The

presentation order of blocks in each run was randomized within each epoch. One epoch consisted of one block from each of 20 conditions and 5 resting blocks (8 sec). For each epoch, 20 unique images per condition were randomly split across 5 scotoma conditions. This procedure was repeated twice and concatenated with a constraint that the same condition cannot appear in two successive blocks. Thus, each of 20 condition blocks were repeated twice per run.

The classic category localizer was 6.9 min (208 TRs) and consisted of four conditions: scenes, faces, objects, and scrambled objects. Ten blocks per condition were acquired within a run. In each condition block (8 sec), four unique images were selected, and one of those images was randomly chosen and repeated twice in a row. Participants performed the one-back repetition task. Each image was presented for 1.1 sec and followed by 500 ms blank. In each run, the block order was randomized within each epoch, which consisted of one block from each condition and one fixation block (8 sec). This procedure was repeated ten times, and the block orders were concatenated across the epochs.

Additionally, the same retinotopy protocol from Experiment 2 was run. All stimuli presentation and the experiment program were produced and controlled by MATLAB R2020b and Psychophysics Toolbox (3.0.17)[67,68].

### Behavioral recognition task

To test whether participants can recognize a basic category of stimuli, a 2-alternative-forced choice (2AFC) was performed inside the scanner during an MPRAGE protocol. Only the face arrays and object arrays with scotomas were tested. Each array was presented for 1.1 sec, which was the same duration used in the main protocol. Then, participants were asked to indicate whether the stimulus was faces or objects, using a response button box.

### fMRI data acquisition

All neuroimaging data were collected at the Harvard Center for Brain Sciences using the bottom half (20 channels) of a 32-channel phased-array head coil with a 3T Siemens Prisma fMRI Scanner. High-resolution T1-weighted anatomical scans were acquired using a 3D multi-echo MPRAGE protocol[69] (176 sagittal slices; FOV = 256 mm; $1 \times 1 \times 1$ mm voxel resolution; gap thickness = 0 mm; TR = 2530 ms; TE = 1.69, 3.55, 5.41, and 7.27 ms; flip angle = 7°). Blood oxygenation level-dependent (BOLD) contrast functional scans were obtained using a gradient echo-planar T2* sequence (87 oblique axial slices acquired at a 25° angle off of the anterior commissure-posterior commissure line; FOV = 211 mm; $1.7 \times 1.7 \times 1.7$ mm voxel resolution; gap thickness = 0 mm; TR = 2000 ms; TE = 30 ms, flip angle = 80°, multiband acceleration factor = 3, in-plane acceleration factor = 2)[70-73].

### fMRI data analysis and preprocessing

The fMRI data were analyzed with BrainVoyager 21.2.0 software (Brain Innovation) with custom Matlab scripting. Preprocessing included slice-time correction, linear trend removal, 3D motion correction, temporal high-pass filtering, and spatial smoothing (4mm FWHM kernel). The data were first aligned to the AC-PC axis, then transformed into the standardized Talairach space (TAL). Three-dimensional models of each participant's cortical surface were generated from the high-resolution T1-weighted anatomical scan using the default segmentation procedures in FreeSurfer. For visualizing activations on inflated brains, the segmented surfaces were imported back into BrainVoyager and inflated using the BrainVoyager surface module. Gray matter masks were defined in the volume based on the Freesurfer cortex segmentation.

A general linear model (GLM) was fit for each participant using BrainVoyager. The design matrix included regressors for each condition block and 6 motion parameters as nuisance regressors. The condition regressors were constructed based on boxcar functions for each condition, convolved with a canonical hemodynamic response function (HRF), and were used to fit voxel-wise time course data with

percent signal change normalization and correction for serial correlations. The beta weights from the GLM were used as measures of activation to each condition for all subsequent analyses.

### Regions of interest (ROIs)

Experiment 2 did not have separate localizer runs. So, we split the main runs into two sets and used the half of runs to localize ROIs and the other half to extract data for subsequent analyses. We defined ROIs separately in each hemisphere in each participant, using condition contrasts implemented in subject-specific GLMs. Three scene-selective areas were defined using [Postcard Scenes−Faces] contrast ($p < 0.0001$). Specifically, the PPA was defined by locating the cluster between posterior parahippocampal gyrus and lingual gyrus, the RSC was defined by locating the cluster near the posterior cingulate cortex, and the OPA was defined by locating the cluster near transverse occipital sulcus. The FFA was defined using [Faces−Postcard Scene] contrast ($p < 0.0001$). The early visual areas (EVA; V1–V3) were defined manually on inflated brain, based on the contrast of [Horizontal−Vertical] meridians from the retinotopy runs.

In Experiment 3, independent localizer runs were used to define ROIs. We defined the PPA, RSC, and OPA using [Scenes−Faces] contrast ($p < 0.0001$). The FFA was defined using [Faces−Scenes] contrast ($p < 0.001$). The lateral occipital complex (LOC) was defined using [Objects−Scrambled Objects] contrast ($p < 0.0001$). Finally, the early visual areas (EVA; V1–V3) were defined manually on the inflated brain based on the contrast of [Horizontal−Vertical] meridians from the retinotopy runs. All ROIs were defined separately in each hemisphere of each participant.

### Eccentricity preference map

To examine a topographic mapping of the eccentricity map, we calculated a group-level preference map. First, responses to each of 5 levels of eccentricities were extracted in each voxel from single-subject GLMs and then averaged over subjects. For each voxel, a condition showing the highest group-average response was identified as the preferred condition. The degree of preference was computed by taking the response differences between the most preferred condition and the next most preferred condition. For visualization, we colored each voxel with a color hue corresponding to the preferred condition, with a color intensity reflecting the degree of preference. The resulting preference map was projected onto the cortical surface of a sample participant. The same preference mapping procedures were used to generate individual subject preference mapping as well.

### Reporting summary

Further information on research design is available in the Nature Portfolio Reporting Summary linked to this article.

## Data availability

Preprocessed fMRI data, design matrices, ROI masks, and all stimuli before the image transformation are available in the Open Science Framework repository (https://osf.io/5hsbv). Source data are provided with this paper.

## Code availability

Image transformation scripts are available on Github (https://github.com/jpark203/FullField-ImageWarping; https://doi.org/10.5281/zenodo.11113136) and on the method website (https://jpark203.github.io/fullfield-neuroimaging).

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

## Acknowledgements

Research reported in this study was supported by the Harvard Brain Science Initiative Postdoc Pioneers Grant awarded to J.P. and the National Eye Institute of the National Institutes of Health under Award Number R21EY031867 awarded to T.K. The content is solely the responsibility of the authors and does not necessarily represent the official views of the National Institutes of Health. This research was carried out at the Harvard Center for Brain Science and involved the use of instrumentation supported by the NIH Shared Instrumentation Grant Program (S10OD020039). We acknowledge the University of Minnesota Center for Magnetic Resonance Research for the use of the multiband-EPI pulse sequences. We also thank MCB Graphics at Harvard University for their assistance with the graphics in figures.

## Author contributions

J.P. and T.K. designed the research, interpreted data, and wrote the paper. E.S. designed and constructed the physical apparatus for image projection and developed the computational warping algorithm. J.P. and J.S. collected data. R.M. performed an MRI signal quality assessment. All data preprocessing and experimental analyses were performed by J.P.

## Competing interests

The authors declare no competing interests.
