## [Peer Review File · Nature Communications]

Immersive scene representation in human visual cortex with
ultra-wide angle neuroimagingReviewer #1 (Remarks to the Author):

This paper is a technical tour de force that opens up interesting new lines of research on how the human brain processes natural scenes. Decades of research have used functional MRI to map and characterize human visual cortex; however, the vast majority of these studies have been constrained to small viewing angles (typically 15-30 deg), well short of the 220 deg view in natural vision. By overcoming this problem, the authors are able to perform some of the first systematic investigations into human peripheral vision and lay the way for more naturalistic neuroimaging studies.

The approach is clever and well described. The team has gone beyond expectations in making information openly available to other groups who wish to employ the approach – Kudos! The fMRI experiments provide an initial foray into the sorts of questions that can be addressed with wide-field projection. The experiments are well motivated and well executed. The paper is easy to read. The insight that scene-selective regions are close to regions that represent the far periphery is important.

In my view, the paper could be published as is, though I suggest some minor revisions to make the methods clearer and I provide some additional points for consideration at the authors' discretion.

Minor comments

The methods describe one important caveat that should be clearer and more obvious in the main manuscript: "approximately 1/3 of the Y pixels of the projector fall onto the screen, limiting our vertical resolution to 284 pixels rather than the native 768." (Lines 507-509). So if I understand, the actual aspect ratio was 1024 x 284 (~3.6:1). However, the visual angles are less clear. The manuscript states: "The total screen extends over 2.3 degrees. The ultra-long throw lens we used, ratio=9 creates an image that spans 4.8 degrees." (Lines 509-510). Does this mean the projected image was 175 deg wide x only 2.3 deg high? That seems unlikely and a big limitation so maybe I don't understand. Or does it match the aspect ratio of 3.6:1, which would be more like 175 x 48 deg? This must be clarified. Were the vertical parts of the image that did not land on the screen masked or were they visible on the bore? If the actual aspect ratio is not what is represented in the figures (retinotopy and scene stimuli), the figures should be adjusted to reflect what subjects saw, not what the projector emitted. This limitation should be addressed in the discussion.

There were pragmatic questions I wondered about in terms of actually using such a setup in a shared MRI facility that requires setting up and taking down apparatus for different experiments. How much of a nightmare is the setup? How long does it take? Does it have to be recalibrated each time?

Line 647: "All neuroimaging data were collected at the Harvard Center for Brain Sciences using a 32-channel phased-array head coil with a 3T Siemens Prisma fMRI Scanner." Specify that you only used the bottom half of the coil, which likely has 20 channels (e.g., Freud et al, 2018, Cortex, PMID: 28431740).

Line 190: "The retrosplenial cortex (RSC) immediately abuts the Peripheral-POS region". In Figure 3 it appears to overlap, not just abut. Can you clarify?

Methods/Apparatus: Was there any rationale behind selecting the curvature of the screen? For example, it could be shaped to keep points along the y axis equidistant from the eye and in focus. I am aware this might not matter that much because acuity is so poor in the periphery.

Was there any task (e.g., one back) or was it purely passive viewing?

I applaud the move toward "investigating more naturalistic, immersive scene perception inside a scanner." Indeed, the authors might even go further in developing the "Current limit and future direction" (shouldn't that be "directions"?), though I'll consider these "take it or leave it" suggestions.

This section may benefit from a deeper consideration of what peripheral vision is used for, such as navigation, locomotion, optic flow, obstacle avoidance, attention/search, guiding gaze, visually monitoring body state especially in the lower visual field). A good reference would be Vater, Wolfe & Rosenholtz (2022 Psychon Bull Rev, PMID: 35581490). This also raises an important caveat with the present work. Here the task appears to be passive viewing (or a one-back task?). This is a reasonable first step, but perhaps it's not surprising that the peripheral effects were strongest in early retinotopic cortex (V1-V2-V3). Perhaps the value of wide projection is the ability to study peripheral visual function, which has been largely neglected in psychophysics (with exceptions like Rosenholtz) and almost entirely neglect in imaging. Another limitation of the retinotopy studies here is that they only investigated eccentricity but not polar angle. Given that Sereno/Pitzalis and colleagues (refs in paper) found V6A by extending the visual field to 55 deg, could an advanced analysis of retinotopy reveal other new discoveries?

To really bring the real world into the scanner, one would want to use the present methods with 3D projector (such as the PROPixx MRI), though there may be challenges dealing with issues like the vergence-accommodation conflict (one would want scenes ~at the plane of fixation) and horopter. I think there is also room to veridically render scenes in which retinal sizes match what would be seen in the real world. Here it sounds like there were approximate attempts to do so. My lab has been working on this with standard projection and I'd be happy to discuss further if the authors are interested in doing this in the future.

Jody Culham

Reviewer #2 (Remarks to the Author):

The authors present a method for display of wide-angle stimuli in fMRI experiments. They use this method to investigate the representation of visual space (specifically the far periphery) and image contents in scene-selective areas, and report that both scene- and face-selective regions maintain their selectivity for scenes and faces substantially into the periphery,. They also find that scene selective areas do not show increasing responses out to the far periphery, and offer an update to the peripheral bias hypothesis.

The method is technically impressive, sound, and valuable, and the scientific results are interesting. I think this is a valuable contribution and the only issues I see are relatively minor. Broadly, these relate to a few things to better quantify, and to changes in the writing.

To investigate the effects of the removal of the top head coil on MRI and fMRI signal, the authors compute T1 image intensity and tSNR. Full brain images of each are shown as qualitative evidence for similar signals in posterior cortex with or without the top coil. To provide a direct assessment of signal in areas relevant to the study, the authors should quantify these effects in place-selective regions of interest.

Some technical details of the projection setup were a bit difficult to follow. The diagram in Supplementary Figure 6 should include units for the numbers. A more detailed diagram of the projection onto the screen - with dimensions given in pixels, inches, and visual angle, with reflection angles labeled appropriately - would be very useful. For example, it was difficult to follow specifically how the vertical resolution of the screen was diminished by the projection setup. In the discussion the authors explain fairly clearly that the geometrical constraints of the room forced the vertical extent of the projected image to be spread over a larger vertical dimension than the screen in the magnet. This detail does not seem to be present in the methods or the diagram of the projection in Figure 1 (it should be). The authors write: "As a result, approximately 1/3 of the Y pixels of the projector fall onto the screen, limiting our vertical resolution to 284 pixels rather than the native 768. The total screen extends over 2.3 degrees."

I am not sure how to understand that last sentence - is 2.3 degrees the total vertical extent in visual angle? (this seems unlikely as this would be very small). Also: what were the dimensions of

the rendered image fed to the warping algorithm, and what were the dimensions of the output of the warping algorithm? Were there buffer zones at the top and bottom of the output of the warping, similar to the old letterbox format for movies? The resolution - at 2-4 pixels per degree - also seems to be a fairly substantial limit of the projection method as implemented. The authors point out that this could be substantially helped with a higher resolution projector. It would be useful to provide clear transformations for common screen resolutions (1024 x 768, 1920 x 1080, 4k) into pixels per degree in different parts of the visual field.

I would like to note that these are challenging problems to solve and that the authors have done very good work to solve them; I would just encourage a bit more detail about the limits and/or remaining challenges. Since the method is a main point of the paper, a thorough description of it seems important.

A few more wording / discussion issues:

The authors describe the cortex posterior to the POS as having "retinotopic preference (regardless of content)"; this may be too strong. Medial parietal areas close to the POS have been investigated in multiple studies, and have been found to be selective for motion, particularly self motion. The authors should provide appropriate caveats in their description of their results (e.g. retinotopic preference that was uniform across the types of image variation shown in this study) and cite and discuss relevant work on V6 (e.g. Pitzalis et al, 2013, Selectivity to Translational Egomotion in Human Brain Motion Areas; Pitzalis et al, 2013, The functional role of the medial motion area V6; Sereno, Pitzalis, & Martinez, 2001, Mapping of Contralateral Space in Retinotopic Coordinates by a Parietal Cortical Area in Humans; the authors already cite but do not say much about Pitzalis, Sereno, et al, 2006, Wide-field retinotopy defines human cortical visual area V6)

On peripheral bias: Past studies that suggested a peripheral bias in scene-selective areas have looked at a smaller range of eccentricities due to limits on the size of displays - which this paper overcomes. It would be useful to be a bit clearer about what eccentricity range has previously been tested to support the eccentricity bias hypothesis and what is new here. To the best of my recollection, the present results are consistent with past eccentricity bias results, in that responses to eccentricities do increase out to ~20 degrees of eccentricity. I don't think that the range beyond that that has previously been tested.

The authors do not report tracking their participants' gaze. It does not seem reasonable to ask that the authors implement gaze tracking alongside the considerable and complicated methodological improvements that they do report. However, the authors should discuss the potential for eye tracking along with the projection system they have implemented, and some potential caveats to their scientific conclusions due to the absence of gaze tracking.

On the potential for eye tracking: Can the authors see a way by which it would be possible in principle to implement eye tracking with this projection system? Or do physical constraints (i.e. the lack of a mirror over the subjects' eyes) that rule it out?

On caveats to the results: Is it possible that the face responses in the far periphery may be due to intermittent peeks at the stimuli? Do the authors have reason to believe that these participants were excellent at fixating?

These questions should be addressed in the manuscript

I think that all of these issues can be fixed with some straightforward changes to the writing.

A few typos and other small things:

Supp figure 1: "poster part of the brain" -> posterior

Figure 2: "Individual brain maps from six participants also show a consistent pattern of results." - There seem to be nine participants shown

Asterisks for significant differences between bars in Fig 6 would be useful

Figure 9 caption: "This flatten view" -> This flattened view

Reviewer #3 (Remarks to the Author):

Park et al. report an exciting innovation: an ultra-wide field-of-view visual presentation system for fMRI setups. Typically, only 15-20 degrees of our visual field is stimulated during an fMRI experiment, due to the limitations of traditional projector setups. In contrast, Park et al.'s presentation setup unlocks the ability to present visual stimuli up to 175 visual degrees, enabling studies of the visual periphery. To do this, the authors have designed a system of mirrors to project the image onto a curved screen on the top of the bore of the magnet, and a warping algorithm to correct the visual image. They also must remove the top coil, limiting SNR of more anterior regions of the brain. The authors provide an exciting resource for the field: the design files and specifications of the system that would enable other researchers to methodologically reproduce their innovation and use it to answer new questions.

Using this new technique, the authors tackle two big-picture questions: 1) what areas of the visual system process scene content, when it is naturalistically displayed (i.e. full field), and 2) how is activity in the traditionally-defined "scene-selective areas" of the brain (PPA, OPA, and RSC) modulated by the presentation of various types of content (i.e., scenes, faces, objects) at various positions in the visual field (i.e. central to far peripheral)? They reach two conclusions: 1) no evidence for new scene-selective peripherally-biased cortex beyond the classic areas (PPA, OPA, and RSC), and 2) classic scene areas are modulated by scene content, irrespective of stimulus eccentricity. In this sense, these findings contribute to our understanding of the nature of scene processing in the brain: scene areas are likely tuned to higher-order image statistics that define scene content, and do not any visual content in the far periphery as "a scene". These findings contribute to theories of the functional organization of visual cortex by revealing, for the first time, how visual areas respond to content in the far periphery. Below, I describe some significant concerns related to the strength and significance of the conclusions that can be drawn from these studies, and share questions related to the impact of the inevitable tSNR drop-off (given no top-coil) for the interpreting the reported results.

Major concerns:

1. The knowledge gap motivating Exp. 2 is whether a full-field visual display could be used to might reveal new regions of the brain that participate in scene analysis beyond the known scene-selective areas ("Will the immersive full-field scenes recruit additional brain regions?"). However, to my eye, the authors don't nail this question. I think that the authors want to know whether there are cortical areas that are dedicated to scene processing (scenes > other categories) beyond the current known scene areas (beyond PPA, OPA, RSC). What they test is whether there are regions of the brain that respond more to full-field scenes > post-card scenes. This contrast reveals an interesting patch of cortex along the POS, which hasn't previously been described. But the authors do not test whether this region is, in fact, dedicated to scene processing (scenes > other categories). As the authors note, this area may simply be involved in stimulus-general far-peripheral information processing. It strikes me that the claim the authors want to make – that there is no region of the brain that extends beyond the current scene areas (PPA, OPA, and RSC) – would require four contrasts (full-field scenes – full-field faces > post-card scenes - post-card faces). Without such a set of conditions, I don't think that the authors can test the question at hand: whether the POS area (or any other brain areas beyond PPA, OPA, and RSC) is involved in far-peripheral scene processing (as compared with stimulus-general far-peripheral information processing). As a result, I am hesitant to accept the conclusion that "The whole-brain contrasts did not show clear evidence for a new scene region, or more extensively activated cortical surface area from the classic scene regions".

2. Relatedly, if the authors want to test whether immersive, full-field scene presentation (175 degrees) recruits additional cortex beyond the classically defined scene areas, it would be important to match the size of stimuli in the baseline condition (post-card scenes) to the stimulus

size that is traditionally used to define scene areas (10-15 degrees). Instead, the baseline condition here is (44 degrees). So, again, I'm hesitant to accept the conclusion that there is no evidence for "more extensively activated cortical surface area from the classic scene areas". How do we know that the baseline condition used in this study (44 degrees) isn't already activating more extensive cortical surface area than would be seen with the smaller stimuli used in previous studies (10-15 degrees)?

3. Another major knowledge gap tackled by this paper (Exp. 3) is whether areas of the brain that process scene stimuli are preferentially activated by scene content (irrespective of visual field position), or visual field position (irrespective of scene content). The authors show that activity in scene regions is modulated dominantly by image content: scene areas show content-preferences for scene stimuli (as compared with faces or objects) at all eccentricities. I think this question is very interesting, and the results here are strong. However, it is unclear to me how this work relates to previous work on how the degree of content-selectivity for category-specific regions changes as a function of eccentricity. Does this finding simply extend prior work exploring how the content preferences of PPA and FFA change as a function of eccentricity, now expanding our these findings into the far periphery? I would suggest clarifying how the findings reported here go beyond previous exploration of how category-selectivity is modulated by eccentricity.

4. The authors remove the top coil to make this technique work and show tSNR maps on a few slices in the supplemental. However, it is difficult to tell how much tSNR suffers, especially in the ROIs of interest, without a comparison with and without the top coil. Could the authors plot tSNR in the ROIs of interest with and without the top coil?

5. The conceptual model presented in Figure 9 is interesting. The authors suggest that the three scene areas might be connected anatomically by a patch of cortex that is sensitive to peripheral visual stimulation. However, I am having trouble wrapping my head around how this model would incorporate other category-selective areas outside of scenes. It might also be the case that the unwrapped brain would go through other retinotopic maps and/or category-selective areas "on the way" from PPA to OPA. Does it?

Reviewer 1

This paper is a technical tour de force that opens up interesting new lines of research on how the human brain processes natural scenes. Decades of research have used functional MRI to map and characterize human visual cortex; however, the vast majority of these studies have been constrained to small viewing angles (typically 15-30 deg), well short of the 220 deg view in natural vision. By overcoming this problem, the authors are able to perform some of the first systematic investigations into human peripheral vision and lay the way for more naturalistic neuroimaging studies.

The approach is clever and well described. The team has gone beyond expectations in making information openly available to other groups who wish to employ the approach – Kudos! The fMRI experiments provide an initial foray into the sorts of questions that can be addressed with wide-field projection. The experiments are well motivated and well executed. The paper is easy to read. The insight that scene-selective regions are close to regions that represent the far periphery is important.

In my view, the paper could be published as is, though I suggest some minor revisions to make the methods clearer and I provide some additional points for consideration at the authors' discretion.

Minor comments:

- The methods describe one important caveat that should be clearer and more obvious in the main manuscript: “approximately 1/3 of the Y pixels of the projector fall onto the screen, limiting our vertical resolution to 284 pixels rather than the native 768.” (Lines 507-509). So if I understand, the actual aspect ratio was 1024 x 284 (~3.6:1). However, the visual angles are less clear. The manuscript states: “The total screen extends over 2.3 degrees. The ultra-long throw lens we used, ratio=9 creates an image that spans 4.8 degrees.” (Lines 509-510). Does this mean the projected image was 175 deg wide x only 2.3 deg high? That seems unlikely and a big limitation so maybe I don't understand. Or does it match the aspect ratio of 3.6:1, which would be more like 175 x 48 deg? This must be clarified.

Thanks for raising these confusions (we also confused Reviewer 2 in the same way). We have revised the manuscript to clarify these points, and added a new supplemental figure. For example, in Line 509 that you reference above, we were not referring to the visual angle for a participant, but the vertical angular extent of projected image that can be on the screen. We clarified this in the manuscript (Line 546 – 560; Supplementary Fig. 8a).

More generally, our original stimuli had the aspect ratio of 4:3 (1024 x 768 pixels). The cylindrical display surface of our screen has the same 4:3 aspect ratio as it measured 58.5cm (arc length) x 44cm (linear). Our image warping algorithm allows us to project the images onto the cylindrical screen without stretch or distortion; similar to the real-world action of covering a cylindrical surface with wall paper.

The visual angle of the entire screen is 175 x 111.5 deg, when the distance from the eyes to the center of screen is 15cm. However, the distance to the screen varies based on the participant's head size and cushion, so we updated our manuscript to report a potential range of visual angle depending on the viewing distance (13.5 – 16.5cm), resulting in 168 – 182 deg in width, 106 – 117 deg in height.

We have made these clarifications in the revised manuscript (also as per Reviewer 2's same comments).

- Were the vertical parts of the image that did not land on the screen masked or were they visible on the bore?

Pixels that did not intercept our display screen were set to black (added in line 549).

- If the actual aspect ratio is not what is represented in the figures (retinotopy and scene stimuli), the figures should be adjusted to reflect what subjects saw, not what the projector emitted. This limitation should be addressed in the discussion.

The original aspect ratio was preserved on the curved screen via image transformation. Estimates of stimulus angular extent have been calculated for a single hypothetical viewpoint. Perspective estimates did not take into account subject variability or binocularity. We have clarified these points in the manuscript (Line 565 - 569).

- There were pragmatic questions I wondered about in terms of actually using such a setup in a shared MRI facility that requires setting up and taking down apparatus for different experiments. How much of a nightmare is the setup? How long does it take? Does it have to be recalibrated each time?

Good questions!

We allocated about 15-20 minutes of additional time for the following steps.

- 1) dropping down the upper mirror using a hinge*
- 2) replacing the top head coil with a custom screen*
- 3) adjusting the focus (+,-) of projector as the distance to screen is changed*
- 4) using a reference image (warped), checking the position of projected image on the screen, and if necessary, making slight adjustments on the projector position and/or the mirror angle.*
- 5) further adjusting the focus after a participant is in position*

The steps 1-3 are quick and straightforward, whereas steps 4-5 can be much more time consuming if the adjustments are needed. We have added this to the methods section.

- Line 647: “All neuroimaging data were collected at the Harvard Center for Brain Sciences using a 32-channel phased-array head coil with a 3T Siemens Prisma fMRI Scanner.” Specify that you only used the bottom half of the coil, which likely has 20 channels (e.g., Freud et al, 2018, Cortex, PMID: 28431740).

Thank you for pointing this out. We now specify that we neuroimaging data were collected with used the base (20-channels) of a standard 32-channel head coil.

- Line 190: “The retrosplenial cortex (RSC) immediately abuts the Peripheral-POS region”. In Figure 3 it appears to overlap, not just abut. Can you clarify?

For each participant, we defined RSC and Peripheral-POS independently, using different contrasts and different data sets. To examine the spatial overlap between these two regions, we quantified the proportion of Peripheral-POS voxels that were also defined as RSC.

There were some individual variations (0 - 24%), but on average, the overlap was about 5.6% of the Peripheral-POS voxels. For the simplicity, in our subsequent ROI analyses (e.g., Fig.4), we removed the overlapping voxels from the Peripheral-POS. We updated the manuscript and clarified our procedure (Line 183 – 185).

- Methods/Apparatus: Was there any rationale behind selecting the curvature of the screen? For example, it could be shaped to keep points along the y axis equidistant from the eye and in focus. I am aware this might not matter that much because acuity is so poor in the periphery.

The radius of the cylindrical screen is 27.94 cm (11"). The screen was made as large as possible while remaining rigidly supported and still fitting inside the MRI bore (~12" radius). The 1" difference allows for the supporting ribs of the hull and bit of clearance when moving in and out of the bore. We added this rationale to the methods section (Line 508 – 511). We didn't think about setting the curvature to keeping points equidistance from the eyes—that's smart, and perhaps will be in the screen-design-2.0. Thanks!

- Was there any task (e.g., one back) or was it purely passive viewing?

Participants performed a fixation dot color detection task for the retinotopic protocols. For the rest of experiments, they performed a one-back repetition detection. We now include these details in Results as well (Line 105-107).

I applaud the move toward “investigating more naturalistic, immersive scene perception inside a scanner.” Indeed, the authors might even go further in developing the “Current limit and future direction” (shouldn’t that be “directions”?) section, though I’ll consider these “take it or leave it” suggestions.

- This section may benefit from a deeper consideration of what peripheral vision is used for, such as navigation, locomotion, optic flow, obstacle avoidance, attention/search, guiding gaze, visually monitoring body state especially in the lower visual field). A good reference would be Vater, Wolfe & Rosenholtz (2022 Psychon Bull Rev, PMID: 35581490). This also raises an important caveat with the present work. Here the task appears to be passive viewing (or a one-back task?). This is a reasonable first step, but perhaps it’s not surprising that the peripheral effects were strongest in early retinotopic cortex (V1-V2-V3). Perhaps the value of wide projection is the ability to study peripheral visual function, which has been largely neglected in psychophysics (with exceptions like Rosenholtz) and almost entirely neglect in imaging.

We agree that there are a lot of interesting research directions that are unlocked by the full-field neuroimaging. We broadened our discussion on this topic.

- Another limitation of the retinotopy studies here is that they only investigated eccentricity but not polar angle. Given that Sereno/Pitzalis and colleagues (refs in paper) found V6A by extending the visual field to 55 deg, could an advanced analysis of retinotopy reveal other new discoveries?

We agree that there is potential for new discoveries in the retinotopic areas. Running a full-field pRF study is one of the future experiments we think would be most valuable next.

- To really bring the real world into the scanner, one would want to use the present methods with 3D projector (such as the PROPixx MRI), though there may be challenges dealing with issues like the vergence-accommodation conflict (one would want scenes ~at the plane of fixation) and horopter. I think there is also room to veridically render scenes in which retinal sizes match what would be seen in the real world. Here it sounds like there were approximate attempts to do so. My lab has been working on this with standard projection and I’d be happy to discuss further if the authors are interested in doing this in the future.

Jody Culham

That sounds really exciting! Let's absolutely discuss more. We're hoping to get a community of interested people to develop and advance this method. If/when you're ready you can sign up here: <https://jpark203.github.io/fullfield-neuroimaging>.

Reviewer 2

The authors present a method for display of wide-angle stimuli in fMRI experiments. They use this method to investigate the representation of visual space (specifically the far periphery) and image contents in scene-selective areas, and report that both scene- and face-selective regions maintain their selectivity for scenes and faces substantially into the periphery,. They also find that scene selective areas do not show increasing responses out to the far periphery, and offer an update to the peripheral bias hypothesis.

The method is technically impressive, sound, and valuable, and the scientific results are interesting. I think this is a valuable contribution and the only issues I see are relatively minor. Broadly, these relate to a few things to better quantify, and to changes in the writing.

- To investigate the effects of the removal of the top head coil on MRI and fMRI signal, the authors compute T1 image intensity and tSNR. Full brain images of each are shown as qualitative evidence for similar signals in posterior cortex with or without the top coil. To provide a direct assessment of signal in areas relevant to the study, the authors should quantify these effects in place-selective regions of interest.

We now include tSNR computed within each ROI, with and without the top head-coil in the Supplement Information (Fig S2). There was clear decrease of tSNR in all of main ROIs when the top head-coil was removed. However, it is noteworthy that overall tSNR values without the top head-coil are still in the high range (> 100), due to spatial smoothing, re-slicing, and averaging across voxels within an ROI, which are the identical steps we took for the actual experiment data. We have also clarified our discussion of loss of tSNR in the main manuscript accordingly (Line 453-456).

- Some technical details of the projection setup were a bit difficult to follow. The diagram in Supplementary Figure 6 should include units for the numbers. A more detailed diagram of the

projection onto the screen - with dimensions given in pixels, inches, and visual angle, with reflection angles labeled appropriately - would be very useful.

Thank you for the suggestion. We updated our figure (Supplementary Fig. 8).

- For example, it was difficult to follow specifically how the vertical resolution of the screen was diminished by the projection setup. In the discussion the authors explain fairly clearly that the geometrical constraints of the room forced the vertical extent of the projected image to be spread over a larger vertical dimension than the screen in the magnet. This detail does not seem to be present in the methods or the diagram of the projection in Figure 1 (it should be).

We updated our methods section and now provide an additional figure to further clarify this (Supplementary Fig. 3, Supplementary Fig. 8).

- The authors write: "As a result, approximately 1/3 of the Y pixels of the projector fall onto the screen, limiting our vertical resolution to 284 pixels rather than the native 768. The total screen extends over 2.3 degrees." I am not sure how to understand that last sentence - is 2.3 degrees the total vertical extent in visual angle? (this seems unlikely as this would be very small).

We acknowledge that description of this part was confusing (we confused Reviewer 1 in the same way!). In Line 509, we were not referring to the visual angle for a participant, but the vertical angular extent of projected image that can be on the screen. We have now clarified this in the manuscript (Line 546 – 560; Supplementary Fig. 8a).

- Also: what were the dimensions of the rendered image fed to the warping algorithm, and what were the dimensions of the output of the warping algorithm? Were there buffer zones at the top and bottom of the output of the warping, similar to the old letterbox format for movies?

All our original stimuli had the aspect ratio of 4:3 (1024 x 768 pixels). The cylindrical display surface of our screen has the same 4:3 aspect ratio as it measured 58.5cm (arc length) x 44cm (linear). Our image warping algorithm allows us to project the images onto the cylindrical screen without stretch or distortion; similar to the real world action of pasting a sheet of wallpaper onto a cylindrical wall. We have added these details to the revised manuscript (also as per Reviewer 1's same comments).

- The resolution - at 2-4 pixels per degree - also seems to be a fairly substantial limit of the projection method as implemented. The authors point out that this could be substantially helped with a higher resolution projector. It would be useful to provide clear transformations for common screen resolutions (1024 x 768, 1920 x 1080, 4k) into pixels per degree in different parts of the visual field.

With the current screen, the pixels per degree (px/deg) is 4.6 - 4.9 in width and 2.4 - 2.7 in height. With the 4k projector (maximum 3840 x 2400 pixels), and with the same aspect ratio of 4:3 (3200 x 2400), the scaling factor for resolution would be $2400/768 = 3.125$, resulting in 14.4 - 15.3 px/deg in width and 7.5 - 8.4 px/deg in height. We updated manuscript accordingly and included details on how we computed these (Line 556-560).

I would like to note that these are challenging problems to solve and that the authors have done very good work to solve them; I would just encourage a bit more detail about the limits and/or remaining challenges. Since the method is a main point of the paper, a thorough description of it seems important.

A few more wording / discussion issues:

- The authors describe the cortex posterior to the POS as having "retinotopic preference (regardless of content)"; this may be too strong. Medial parietal areas close to the POS have been investigated in multiple studies, and have been found to be selective for motion, particularly self motion. The authors should provide appropriate caveats in their description of their results (e.g. retinotopic preference that was uniform across the types of image variation shown in this study) and cite and discuss relevant work on V6 (e.g. Pitzalis et al, 2013, Selectivity to Translational Egomotion in Human Brain Motion Areas; Pitzalis et al, 2013, The functional role of the medial motion area V6; Sereno, Pitzalis, & Martinez, 2001, Mapping of Contralateral Space in Retinotopic Coordinates by a Parietal Cortical Area in Humans; the authors already cite but do not say much about Pitzalis, Sereno, et al, 2006, Wide-field retinotopy defines human cortical visual area V6)

Thanks for these references. We now provide more context on the Peripheral-POS region and other relevant areas (e.g., V6, Prostriata, RSC) in the manuscript (Line 378 - 386).

We did not intend to claim that the Peripheral-POS is modulated by retinotopic properties only, regardless of *all* content—we now clarify in the text that ‘content’ here is used to refer to our manipulated variable (intact vs scrambled scene content).

- On peripheral bias: Past studies that suggested a peripheral bias in scene-selective areas have looked at a smaller range of eccentricities due to limits on the size of displays - which this paper overcomes. It would be useful to be a bit clearer about what eccentricity range has previously been tested to support the eccentricity bias hypothesis and what is new here. To the best of my recollection, the present results are consistent with past eccentricity bias results, in that responses

to eccentricities do increase out to ~20 degrees of eccentricity. I don't think that the range beyond that that has previously been tested.

Thank you for the suggestion. As pointed out, the eccentricity beyond 20 deg has not been tested for the peripheral bias. We now explicitly discuss the maximum eccentricity tested in the previous studies (Levy et al., 2001; Hasson et al., 2002; Silson et al., 2015; 2016a; 2016b), and highlight the new insights gained from our study more clearly.

- The authors do not report tracking their participants' gaze. It does not seem reasonable to ask that the authors implement gaze tracking alongside the considerable and complicated methodological improvements that they do report. However, the authors should discuss the potential for eye tracking along with the projection system they have implemented, and some potential caveats to their scientific conclusions due to the absence of gaze tracking.

It is true that the absence of gaze tracking is one of limitations of the current full-field setup. We added this caveat in the Discussion.

- On the potential for eye tracking: Can the authors see a way by which it would be possible in principle to implement eye tracking with this projection system? Or do physical constraints (i.e. the lack of a mirror over the subjects' eyes) that rule it out?

The full-field screen blocks the eye tracker. The standard eye tracker's camera and illuminator are at the end of the bore, so it requires the mirror to "see" the participant's eyes. Since we removed this mirror (and the top head-coil), and the screen is positioned in the mirror's spot, it is not possible to use the standard eye tracker. We think that emerging tools like Deep MREye*, which infer eye-position through the EPI signals of the eyeballs, may be an interesting way to address this going forward.

****Frey, M., Nau, M., & Doeller, C. F. (2021). Magnetic resonance-based eye tracking using deep neural networks. Nature neuroscience, 24(12), 1772-1779.***

- On caveats to the results: Is it possible that the face responses in the far periphery may be due to intermittent peeks at the stimuli? Do the authors have reason to believe that these participants were excellent at fixating?

We thought about this as well. While we cannot completely rule out this possibility, it seems unlikely that the result is merely driven by the eye movements. First, if participants consistently shifted their gaze towards peripheral faces, activation levels at the largest scotoma would likely match those in the no-scotoma condition, as these faces are actually foveally presented. Second, occasional eye movements would likely result in greater variance in the far-periphery condition, which we did not observe. Lastly, it actually requires considerable efforts to move eyes beyond 70 degrees (one direction). We have added these thoughts to the revised manuscript (Line 418 – 426).

These questions should be addressed in the manuscript

I think that all of these issues can be fixed with some straightforward changes to the writing.

A few typos and other small things:

- Supp figure 1: "poster part of the brain" -> posterior
- Figure 2: "Individual brain maps from six participants also show a consistent pattern of results." There seem to be nine participants shown
- Asterisks for significant differences between bars in Fig 6 would be useful
- Figure 9 caption: "This flatten view" -> This flattened view

Thank you for pointing these out. We corrected them in the manuscript.

Reviewer 3

Park et al. report an exciting innovation: an ultra-wide field-of-view visual presentation system for fMRI setups. Typically, only 15-20 degrees of our visual field is stimulated during an fMRI experiment, due to the limitations of traditional projector setups. In contrast, Park et al.'s presentation setup unlocks the ability to present visual stimuli up to 175 visual degrees, enabling studies of the visual periphery. To do this, the authors have designed a system of mirrors to project the image onto a curved screen on the top of the bore of the magnet, and a warping algorithm to correct the visual image. They also must remove the top coil, limiting SNR of more anterior regions of the brain. The authors provide an exciting resource for the field: the design files and specifications of the system that would enable other researchers to methodologically reproduce their innovation and use it to answer new questions.

Using this new technique, the authors tackle two big-picture questions: 1) what areas of the visual system process scene content, when it is naturalistically displayed (i.e. full field), and 2) how is activity in the traditionally-defined "scene-selective areas" of the brain (PPA, OPA, and RSC) modulated by the presentation of various types of content (i.e., scenes, faces, objects) at various positions in the visual field (i.e. central to far peripheral)? They reach two conclusions: 1) no evidence for new scene-selective peripherally-biased cortex beyond the classic areas (PPA, OPA, and RSC), and 2) classic scene areas are modulated by scene content, irrespective of stimulus eccentricity. In this sense, these findings contribute to our understanding of the nature of scene processing in the brain: scene areas are likely tuned to higher-order image statistics that define scene content, and do not any visual content in the far periphery as "a scene". These findings contribute to theories of the functional organization of visual cortex by revealing, for the first time, how visual areas respond to content in the far periphery. Below, I describe some significant concerns related to the strength and significance of the conclusions that can be drawn from these studies, and share questions related to the impact of the inevitable tSNR drop-off (given no top-coil) for the interpreting the reported results.

1. The knowledge gap motivating Exp. 2 is whether a full-field visual display could be used to might reveal new regions of the brain that participate in scene analysis beyond the known scene-selective areas ("Will the immersive full-field scenes recruit additional brain regions"?). However, to my eye, the authors don't nail this question. I think that the authors want to know whether there are cortical areas that are dedicated to scene processing (scenes > other categories) beyond the current known scene areas (beyond PPA, OPA, RSC). What they test is whether there are regions of the brain that respond more to full-field scenes > post-card scenes. This contrast reveals an

interesting patch of cortex along the POS, which hasn't previously been described. But the authors do not test whether this region is, in fact, dedicated to scene processing (scenes > other categories). As the authors note, this area may simply be involved in stimulus-general far-peripheral information processing. It strikes me that the claim the authors want to make – that there is no region of the brain that extends beyond the current scene areas (PPA, OPA, and RSC) – would require four contrasts (full-field scenes – full-field faces > post-card scenes - post-card faces). Without such a set of conditions, I don't think that the authors can test the question at hand: whether the POS area (or any other brain areas beyond PPA, OPA, and RSC) is involved in far-peripheral scene processing (as compared with stimulus-general far-peripheral information processing). As a result, I am hesitant to accept the conclusion that “The whole-brain contrasts did not show clear evidence for a new scene region, or more extensively activated cortical surface area from the classic scene regions”.

We directly tested what the reviewer described, by running a conjunction analysis of [[full-field scene – full-field scrambled] – [postcard scene – postcard scrambled]]. We did not find any areas showing differences to this contrast, except for very weak response differences in the Peripheral-POS. We added this result to a supplement figure (Supplementary Fig. 9).

Further, the contrast map of [full-field intact scene – full-field scrambled scene] revealed that the typically defined ROIs largely encircle the strongest areas of intact vs. scrambled response preferences (Fig. 3A). If there was a new scene region additionally recruited in the full-field scene perception, we would expect to see large response differences in this contrast, but this was not the case, further supporting our claim.

2. Relatedly, if the authors want to test whether immersive, full-field scene presentation (175 degrees) recruits additional cortex beyond the classically defined scene areas, it would be important to match the size of stimuli in the baseline condition (post-card scenes) to the stimulus size that is traditionally used to define scene areas (10-15 degrees). Instead, the baseline condition here is (44 degrees). So, again, I'm hesitant to accept the conclusion that there is no evidence for “more extensively activated cortical surface area from the classic scene areas”. How do we know that the baseline condition used in this study (44 degrees) isn't already activating more extensive cortical surface area than would be seen with the smaller stimuli used in previous studies (10-15 degrees)?

We agree—a stronger baseline would be post-card scenes presented at 10-15 degrees. We acknowledge this limitation in the revised manuscript. We focus on what our data do

allow us to say, which is that the extent of activated cortical surface is not much increased by a relatively dramatic stimulus size increase from 44 to 175 deg. And, we can say that even if there is increasing territory as a function of visual angle, it only increases from 15-44 degrees and not beyond (Line 154 – 159).

We also explored an indirect way to address this baseline issue, by comparing our finding to prior studies of ours, where we defined the same ROIs from ‘standard’ scanning in different participants. We did not find statistical differences in any of the ROIs voxel counts, though there was a hint of a trend that there were actually smaller. In the end we decided not to include these exploratory ROI comparison analyses in the manuscript given that they are indirect, subject to other differences (e.g., the full vs bottom half of the head coil, image resolution differences), and thus not controlled enough to be usefully conclusive.

3. Another major knowledge gap tackled by this paper (Exp. 3) is whether areas of the brain that process scene stimuli are preferentially activated by scene content (irrespective of visual field position), or visual field position (irrespective of scene content). The authors show that activity in scene regions is modulated dominantly by image content: scene areas show content-preferences for scene stimuli (as compared with faces or objects) at all eccentricities. I think this question is very interesting, and the results here are strong. However, it is unclear to me how this work relates to previous work on how the degree of content-selectivity for category-specific regions changes as a function of eccentricity. Does this finding simply extend prior work exploring how the content preferences of PPA and FFA change as a function of eccentricity, now expanding our these findings into the far periphery? I would suggest clarifying how the findings reported here go beyond previous exploration of how category-selectivity is modulated by eccentricity.

In the revised manuscript we have attempted to clarify how our data are situated with respect to prior results. Specifically, our study does not directly compare the central-stimulation only vs. peripheral-stimulation only. Instead, the main focus of using a scotoma paradigm was to ask to what extent the central stimulation is necessary for the content preference—e.g., is there a critical portion of central visual field? A potential answer for the face region was “Yes, a face needs to be present within the xx deg of eccentricity to show selectivity for faces”, based on previous findings that the face region has higher activation for foveal stimulation than peripheral stimulation. However, we found that the content preference (i.e., relatively higher activation than other categories) is maintained even at the far-periphery without the central stimulation.

4. The authors remove the top coil to make this technique work and show tSNR maps on a few slices in the supplemental. However, it is difficult to tell how much tSNR suffers, especially in the ROIs of interest, without a comparison with and without the top coil. Could the authors plot tSNR in the ROIs of interest with and without the top coil?

Yes, we agree that the tSNR in the ROIs is critical information. We conducted this analysis and included the results in the manuscript (Supplementary Fig. 2).

5. The conceptual model presented in Figure 9 is interesting. The authors suggest that the three scene areas might be connected anatomically by a patch of cortex that is sensitive to peripheral visual stimulation. However, I am having trouble wrapping my head around how this model would incorporate other category-selective areas outside of scenes. It might also be the case that the unwrapped brain would go through other retinotopic maps and/or category-selective areas “on the way” from PPA to OPA. Does it?

Indeed, all of the rest of object-selective cortex is mapped in between PPA and OPA! This is depicted well in Hasson et al., 2003. We have clarified this in the manuscript (Line 367-373). We made a little schematic below for you as well, but we did not put this in the revised manuscript. We are working on a broader integrative review that highlights these topographic relationships and some version of this diagram will be featured.

** Hasson, U., Harel, M., Levy, I., & Malach, R. (2003). Large-scale mirror-symmetry organization of human occipito-temporal object areas. Neuron, 37(6), 1027-1041.*

Reviewer #1 (Remarks to the Author):

The authors have fully addressed all the questions in my review.

Regarding your reply to another reviewer, we've been using DeepMREye and it works well. You'll likely have to train it on your own data because the training sets they used had much smaller visual angles. The biggest limitation is that the temporal resolution is limited by the TR, but you could still get some sense of how well participants are fixating.

I look forward to seeing the article in print and to seeing what future studies this cool and clever new approach enables.

Reviewer #2 (Remarks to the Author):

The authors have clarified the technical details that were vague or missing in the previous draft, useful discussion has been added, and the paper is strengthened from an already solid point. However, I would still ask the authors to be clearer about the technical details - and limits - in a few specific ways. It is now clear that the paper presents a trade-off: the method described provides a reduced resolution display over a broader field of view.

This is clear enough from the written methods but still not clear from the main figures. For example, it's not obvious in Figure 1 that the image on the right is a transformation of the image at the same size; it looks as if this could be a representation of how the image appears in the bore. The supplemental figures convey this much more clearly (supplementary figure 3 and the additions to supplementary figure 8 are all very nice, but somewhat buried where they are in the supplement). I would ask that the information from supplementary figure 3 and some of the information from supplementary figure 8 be incorporated into a figure (most likely figure 1) or figures in the main manuscript. All images of projections should be labeled with sizes in pixels and degrees. This would convey much more clearly that the image projected onto the screen in the bore was a subset of the full size, only 828 x 284 pixels. It would also help to convey two other important points: First, that this method results in substantially lower resolution, particularly in the vertical dimension, compared to many standard projection systems. In my experience, MRI projection systems are usually in the range of ~20 pixels per degree, making this system ~10 x lower resolution in the vertical dimension. To my mind, an order of magnitude difference in resolution is important, and bears not just a mention in the methods but clear emphasis. Second, the difference in resolution in horizontal and vertical dimensions is substantial. The vertical dimension is stretched twice the amount that the horizontal dimension is, resulting in nearly twice the resolution in the x dimension compared to the y dimension (4.75 vs 2.5 pixels per degree on average). As I said above, labels for pixels & degrees on all projection images would sufficiently address this.

I don't think that the resolution / FOV tradeoff is a problem for the method, but I do think it needs to be conveyed clearly in the figures such that hasty readers don't come away with an inaccurate understanding of the proposed method.

Aside from the points above, the authors have done a good job of addressing the points that the other reviewers and I raised. This is a good paper and a solid contribution; I applaud the authors' work and look forward to seeing this in press.

Reviewer #3 (Remarks to the Author):

The authors have done a fantastic job with this revision. I have no further comments or concerns.

Reviewer #1 (Remarks to the Author):

The authors have fully addressed all the questions in my review.

Regarding your reply to another reviewer, we've been using DeepMREye and it works well. You'll likely have to train it on your own data because the training sets they used had much smaller visual angles. The biggest limitation is that the temporal resolution is limited by the TR, but you could still get some sense of how well participants are fixating.

I look forward to seeing the article in print and to seeing what future studies this cool and clever new approach enables.

We're glad to hear that. Thank you!

Reviewer #2 (Remarks to the Author):

The authors have clarified the technical details that were vague or missing in the previous draft, useful discussion has been added, and the paper is strengthened from an already solid point. However, I would still ask the authors to be clearer about the technical details - and limits - in a few specific ways. It is now clear that the paper presents a trade-off: the method described provides a reduced resolution display over a broader field of view.

This is clear enough from the written methods but still not clear from the main figures. For example, it's not obvious in Figure 1 that the image on the right is a transformation of the image at the same size; it looks as if this could be a representation of how the image appears in the bore. The supplemental figures convey this much more clearly (supplementary figure 3 and the additions to supplementary figure 8 are all very nice, but somewhat buried where they are in the supplement). I would ask that the information from supplementary figure 3 and some of the information from supplementary figure 8 be incorporated into a figure (most likely figure 1) or figures in the main manuscript. All images of projections should be labeled with sizes in pixels and degrees. This would convey much more clearly that the image projected onto the screen in the bore was a subset of the full size, only 828 x 284 pixels. It would also help to convey two other important points: First, that this method results in substantially lower resolution, particularly in the vertical dimension, compared to many standard projection systems. In my experience, MRI projection systems are usually in the range of ~20 pixels per degree, making this system ~10 x lower resolution in the vertical dimension. To my mind, an order of magnitude difference in resolution is important, and bears not just a mention in the methods but clear emphasis. Second, the difference in resolution in horizontal and vertical dimensions is substantial. The vertical dimension is stretched twice the amount that the horizontal dimension is, resulting in nearly twice the resolution in the x dimension compared to the y dimension (4.75 vs 2.5 pixels per degree on average). As I said above, labels for pixels & degrees on all projection images would sufficiently address this.

I don't think that the resolution / FOV tradeoff is a problem for the method, but I do think it needs to be conveyed clearly in the figures such that hasty readers don't come away with an inaccurate understanding of the proposed method.

Aside from the points above, the authors have done a good job of addressing the points that the other reviewers and I raised. This is a good paper and a solid contribution; I applaud the authors' work and look forward to seeing this in press.

Thank you for the suggestion. We agree that the existing illustration of transformed image could be misleading. We now updated figures by incorporating Supplement Fig 3 to the main text and clearly stating the size of transformed image, both in pixel and in visual angle.

Reviewer #3 (Remarks to the Author):

The authors have done a fantastic job with this revision. I have no further comments or concerns.

Thank you!